# Transient mTOR inhibition rescues 4-1BB CAR-Tregs from tonic signal-induced dysfunction

Baptiste Lamarthée [1,14], Armance Marchal[1,14], Soëli Charbonnier [1,14], Tifanie Blein [1,14], Juliette Leon [2], Emmanuel Martin [3], Lucas Rabaux[1], Katrin Vogt[4], Matthias Titeux[5], Marianne Delville [1,6,7], Hélène Vinçon [1], Emmanuelle Six [1], Nicolas Pallet[8], David Michonneau [6,9], Dany Anglicheau [6,10,11], Christophe Legendre[6,10], Jean-Luc Taupin[6,12], Ivan Nemazanyy [13], Birgit Sawitzki [4], Sylvain Latour[3], Marina Cavazzana [1,6,7], Isabelle André [1] & Julien Zuber [1,6,10 ✉]

The use of chimeric antigen receptor (CAR)-engineered regulatory T cells (Tregs) has emerged as a promising strategy to promote immune tolerance. However, in conventional T cells (Tconvs), CAR expression is often associated with tonic signaling, which can induce CAR-T cell dysfunction. The extent and effects of CAR tonic signaling vary greatly according to the expression intensity and intrinsic properties of the CAR. Here, we show that the 4-1BB CSD-associated tonic signal yields a more dramatic effect in CAR-Tregs than in CAR-Tconvs with respect to activation and proliferation. Compared to CD28 CAR-Tregs, 4-1BB CAR-Tregs exhibit decreased lineage stability and reduced in vivo suppressive capacities. Transient exposure of 4-1BB CAR-Tregs to a Treg stabilizing cocktail, including an mTOR inhibitor and vitamin C, during ex vivo expansion sharply improves their in vivo function and expansion after adoptive transfer. This study demonstrates that the negative effects of 4-1BB tonic signaling in Tregs can be mitigated by transient mTOR inhibition.

[1] Laboratoire de lymphohématopoïèse humaine, INSERM UMR 1163, IHU IMAGINE, Paris, France. [2] Department of Immunology, Harvard Medical School, Boston, MA 02115, USA. [3] Lymphocyte activation and susceptibility to EBV, INSERM UMR 1163, IHU IMAGINE, Paris, France. [4] Department of Immunology, Charité University Hospital, Berlin, Germany. [5] Maladie génétique cutanée, INSERM UMR 1163, IHU IMAGINE, Paris, France. [6] Université de Paris, Paris, France. [7] Service de Biothérapie et Thérapie Génique Clinique, Assistance Publique–Hôpitaux de Paris, Hôpital Necker, Paris, France. [8] Université de Paris, INSERM U1138, Centre de Recherche des Cordeliers, 75006 Paris, France. [9] INSERM U976, Paris, France. [10] Service de Transplantation rénale adulte, Assistance Publique-Hôpitaux de Paris, Hôpital Necker, Paris, France. [11] INSERM U1151, Institut Necker Enfants Malades, Paris, France. [12] Laboratoire d'immunologie et histocompatibilité, Assistance Publique-Hôpitaux de Paris, Hôpital Saint-Louis, Paris, France. [13] Plateforme de Métabolique, Structure Fédérative de Recherche, Necker, INSERM US24/CNRS UMS, 3633 Paris, France. [14] These authors contributed equally: Baptiste Lamarthée, Armance Marchal, Soëli Charbonnier, Tifanie Blein. ✉email: julien.zuber@aphp.fr

Despite significant advances in the prevention of acute graft rejection, long-term attrition of graft function and complications related to long-term immunosuppression remain major concerns in the field of solid organ transplantation. FOXP3-expressing regulatory T cells (Tregs) have been shown to play a key role in mitigating alloimmune responses in mouse models[1] and clinical transplantation[2]. Hence, significant efforts have been made to apply Tregs cell therapy in clinical transplantation over the past decade[3]. However, although experimental transplant models indicate that donor-specific Tregs result in greater prevention of graft rejection than polyclonal Tregs, including in highly relevant humanized mouse models[4,5], large-scale generation of clinical-grade donor-specific Tregs faces significant challenges[6]. These include isolation of the scarce antigen-specific population and its subsequent expansion to produce a sizeable cell product[4].

An attractive alternative would be engineering recipient Tregs with a chimeric antigen receptor (CAR) that redirects their antigen specificity toward a given donor antigen. CARs are engineered receptors that consist of an antigen-binding ectodomain, a hinge region linked to the transmembrane domain, and an intracyctoplasmic domain, commonly composed of one or several costimulation domains (CSDs), and CD3ζ signaling tail. The antigen-binding motif is frequently a single-chain variable fragment (ScFv) that combines the heavy and light chains of an antibody[7]. CARs are referred to as first-, second-, and third-generation depending on the number of incorporated CSDs (none, one or several, respectively). Second generation CARs, including either the CD28 or 4-1BB CSD, have been the most broadly used in the clinic.

Although CAR technology has been successfully translated to clinical oncology and extensively refined to further improve the efficacy-safety balance[7], its use in human Tregs is still an emerging field of investigation. A 2011 study demonstrated the ability to weaponize human Tregs using CAR expression[8], and since then, a few pioneering studies have provided proof-of-concept evidence that the Treg response can be redirected toward a donor mismatched antigen in transplant models. More specifically, human leukocyte antigen (HLA) A2-targeted CAR-Tregs efficiently prevent graft-versus-host disease (GVHD) and skin graft rejection in an HLA-A2-specific manner[9–14]. However, there is still limited information concerning the optimal CAR design and components to optimize Treg suppressive function, stability, metabolism, proliferative capacity and survival. In this respect, most of these pilot studies have used second- generation HLA-A2-targeted CARs incorporating the CD28 CSD based on evidence that the CD28 signal is critical for Treg ontogeny[15,16], and survival[17]. On the other hand, 4-1BB expression is a hallmark of activated human Tregs[18], and *TNFRSF9* (tumor necrosis factor receptor superfamily 9, encoding 4-1BB) belongs to a handful of genes whose epigenetic marks are highly conserved in activated Tregs across species[19]. Emerging data suggest that the 4-1BB CSD could support greater CAR-driven proliferation than its CD28 counterparts, while being associated with reduced Treg suppressive function[10,14,20].

CAR tonic signaling may be defined as unduly and sustained CAR-induced activation of T cells in a ligand-dependent or -independent manner. In the field of oncology, CAR tonic signaling was found to occur to varying degrees in response to most CARs, and has been extensively studied through the prism of its impact on the antitumor response, CAR T-cell survival and differentiation[21]. CAR tonic signaling encompasses a wide variety of underlying mechanisms that vary greatly according to the type of vector and the combination of CAR components, including promoter, scFv, hinge, and transmembrane and costimulatory domains[21]. Ligand-independent CAR tonic signaling typically results from high CAR surface expression, correlated with the strength of the promoter[22–24] and/or from the intrinsic properties of the CARs to self-aggregate[25].

The impact of CAR tonic signaling in CAR-Tregs is largely unknown. In this respect, Tregs exhibit unique features compared to Tconvs, which might influence their susceptibility to constitutive and dysregulated activation. This includes greater baseline TCR tonic activation[26], fine control of Akt/mTOR and MAPK activation[27,28], a stage-dependent role for costimulatory molecules[29], specific metabolic demands[30] and a short in vivo life span[31]. Therefore, we aimed to investigate the effect of HLA-A2-specific CAR tonic signaling on CAR-Treg phenotype, proliferation, transcriptomics, metabolism, signaling and function, according to the type of CSD.

Here, we show that 4-1BB tonic signaling significantly impacts the biology of CAR-Tregs and thereby compromises their suppressive function. Transient mTOR inhibition rescues 4-1BB Tregs from functional impairment.

## Results

**Generation of CD28 and 4-1BB HLA-A2-specific CAR-Tregs.** Six anti-HLA-A2 scFvs were derived from the publicly available sequences of three human anti-HLA-A2/A28 antibodies[32]. To assess their specificity, the 6 scFvs were tagged with a polyhistidine tail and then mixed with 96 beads, each of which was coated with a single HLA class I antigen (Supp Fig. 1a). One scFv was selected based on its exclusive binding to beads coated with HLA-A2 and HLA-A28 antigens (Supp Fig. 1b). To assess the impacts of the selected CSD on CAR-Treg biology and function, two anti-HLA-A2 CAR constructs were generated using the selected scFv, incorporating either the CD28 CSD or the 4-1BB CSD, hereafter refered to as CD28 CAR and 4-1BB CAR, respectively (Fig. 1a). A reporter gene encoding truncated epidermal growth factor receptor (EGFRt)[33] was placed behind thosea asigna virus 2A (T2A, a self-cleaving peptide). Both bicistronic constructs were next incorporated into a pCCL self-inactivating lentiviral vector (LV) behind an elongation factor-1 (EF-1) alpha promoter (Supp Fig. 1c). To demonstrate that antigen specificity was maintained after scFv incorporation into the CAR structure, TCR-deficient J.RT3-T3.5 Jurkat cells were transduced with either of the two HLA-A2-specific CARs and then challenged against a panel of irradiated splenocytes isolated from 10 HLA-typed deceased organ donors. Induction of CD69 expression in CAR-expressing Jurkat cells was strictly restricted to those cocultured with irradiated splenocytes expressing either HLA-A2 antigens or HLA-A28 antigens (Supp Fig. 1d).

Since naive Tregs have been found to be more stable over long-term culture than activated Tregs[34,35], CD4+ CD25Hi CD127- CD45RA+ CD45RO− naive Tregs and CD4+ CD25- CD127+ CD45RA+ CD45RO− naive conventional T cells (Tconvs), which served as controls, were sorted by fluorescence-activated cell sorting (FACS) from cytapheresis products collected in young (<40 years old) healthy HLA-A2/A28-negative donors (Fig. 1b). Tregs and Tconvs were transduced with HLA-A2-targeted CAR-encoding LV vectors after 2 days of polyclonal activation with anti-CD3/CD28 beads. EGFRt-expressing transduced T cells were sorted by FACS and restimulated on day 11 before further in vitro (day 16) and in vivo (day 18) assessment (Fig. 1c). To further evaluate HLA-A2-specific activation of CAR-engineered primary T cells, CAR-Tregs were collected on day 10 and, 72 h after bead separation, were restimulated with HLA-A2/A28-positive or HLA-A2/A28-negative irradiated splenocytes. CD69 (Fig. 1d) and GARP (Supp Fig. 2) expression was strongly induced in an HLA-A2-specific manner by EGFR-expressing Tregs but not by untransduced EGFR-negative Tregs. These results indicate that both vectors were efficiently expressed on the cell surface of

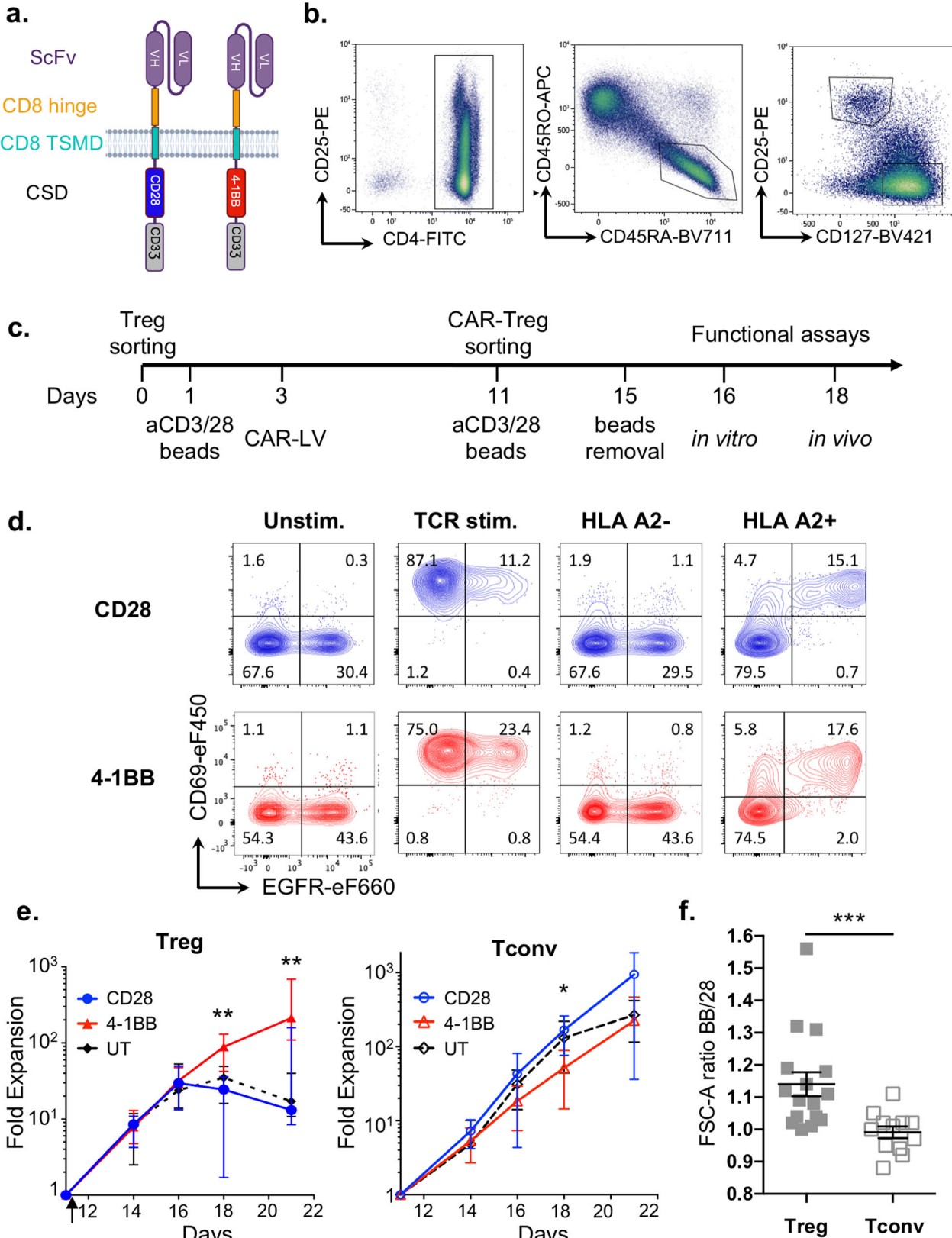

human Tregs and that CAR stimulation was able to induce antigen-specific activation of CAR-expressing human Tregs.

**Treg biology is highly sensitive to 4-1BB CSD-induced tonic signaling.** In line with a recent report[14], we observed that the

proliferative capacity of CAR-Tregs varied greatly between CD28 CAR-Tregs and 4-1BB CAR-Tregs. Indeed, 4-1BB CAR-Tregs achieved 20-fold greater expansion than their CD28 counterparts at day 21. Furthermore, 4-1BB-Treg expansion did not plateau even after 3 weeks of culture, whereas other Treg populations stopped proliferating shortly after the second

**Fig. 1 CD28 and 4-1BB CAR-Treg production. a** CD28 and 4-1BB CAR structures. **b** Gating strategies for CD4+ CD45RA+ CD45RO- CD25hi CD127low naive Tregs and CD4+ CD45RA+ CD45RO- CD25low CD127hi Tconvs. **c** Process timeline for CD28 or 4-1BB CAR-T-cell production. T cells were sorted, stimulated with anti-CD3/CD28 beads, transduced with CAR-encoding lentiviruses, expanded, and sorted based on reporter gene expression, followed by activation with anti-CD3/CD28 beads and expansion for up to 21 days. **d** On day 11, CD28 and 4-1BB CAR-Tregs were incubated with or without anti-CD3/CD28 beads (TCR stim.) or HLA-A2-positive or HLA-A2-negative irradiated splenocytes, and CD69 and EGFR expression was measured by flow cytometry. Contour plots are representative of two experiments. **e** Fold expansion of untransduced Tregs and CAR-Tregs (left panel) or their respective Tconv populations (right panel) from TCR restimulation with anti-CD3/CD28 beads on day 11 (arrow) to day 21 of culture. The median +/− interquartile range (IQR) is shown. A total of 16 independent experiments are shown including the following cell lines: CD28 CAR-Tregs (blue full circle, n = 10), 4-1BB CAR-Tregs (red full triangle, n = 11), untransduced (UT) Tregs (black full diamond, n = 8), CD28 CAR-Tconvs (blue open circle, n = 8), 4-1BB CAR-Tconvs (red open triangle, n = 9), UT Tconvs (black open diamond, n = 8). Two-tailed Mann–Whitney test comparing 4-1BB and CD28 CAR-Tregs indicates significant differences at day 18 (p = 0.0079) and day 21 (p = 0.0012). **f** FSC-A ratios of 4-1BB/CD28 CAR-Tregs and CAR-Tconvs are calculated at the time of CAR-positive cell sorting (D11). Data are depicted as mean +/− SEM and compared using two-tailed Mann–Whitney test. Statistical significance is found between Tconv and Treg populations: ***p = 0.0004. N = 16 donors for Tregs and n = 12 for Tconvs, from 16 independent experiments. Color code: gray full square = ratio 4-1BB/28 CAR-Tregs, gray square = ratio 4-1BB/28 CAR-Tconvs.

CD3/CD28 stimulation (Fig. 1e, left panel). Consistent with this heightened proliferation, 4-1BB CAR-Tregs exhibited a blastic phenotype throughout cell culture, as observed by flow-cytometry forward scatter patterns (Fig. 1f). These results were Treg specific, as 4-1BB CAR-Tconvs and CD28 CAR-Tconvs showed similar growth profiles and cell sizes (Fig. 1e, f).

However, in contrast to previous studies, we found that the 4-1BB CSD had a positive effect on Treg proliferation when CAR-Tregs were stimulated with anti-CD3/CD28 beads in a CAR-independent manner. This indicated that 4-1BB-driven enhanced proliferation was constitutive and ligand independent and could therefore be ascribed to tonic signaling[21–23]. We next sought to further investigate the underlying mechanisms and consequences of 4-1BB CAR tonic signaling in Tregs at day 16, when 4-1BB CAR-Tregs were still expanding, while the CD28 CAR-Tregs had started to plateau (Fig. 1e).

Sustained expression of HLA class II was recently identified in CAR-Tconvs as a hallmark feature of the 4-1BB CSD tonic signal, which was distinct from the tonic CD3ζ signaling signature[36]. Strikingly, HLA-DR was strongly upregulated in 4-1BB CAR-Tregs compared to CD28 CAR-Tregs and to a much greater extent than in their Tconv counterparts (Fig. 2a). As the metabolic checkpoint kinase mTORC1 is crucial for supporting the proliferative capacities of Tregs[37,38], and associated anabolic demands, we examined mTORC1 activity, through the phosphorylation of S6 (pS6) in 4-1BB CAR-Tregs and CD28 CAR-Tregs. In contrast to CD28 CAR-Tregs and untransduced Tregs, 4-1BB CAR-Tregs displayed high expression of pS6 five days after TCR stimulation, whereas mTORC1 activity was similar across Tconv populations, irrespective of CAR CSD (Fig. 2b). In line with this, the comparison of metabolic profiles across lineages and CAR CSDs showed that the 4-1BB-CAR tonic signal had a far greater impact on Treg (but not Tconv) metabolism than the CD28-CAR tonic signal (Fig. 2c). We next determined the 20 metabolites, detected across all cell populations, whose levels were the most significantly increased or decreased in 4-1BB-CAR-Tregs compared to untransduced Tregs (Supp Fig. 3a). Notably, this top 20 metabolite-based profile was remarkably consistent across independent experiments, in a lineage- (Treg vs. Tconv) and CAR CSD-specific manner (Fig. 2d and Supp Fig. 3b). More specifically, glutamine (GLN, Supp Fig. 3b) and glutamate (not shown) levels consistently exhibited a decrease in 4-1BB-CAR-Tregs, in contrast to a sharp increase in α-ketoglutarate (aKG) levels (Fig. 3d). Along with the high expression of the glutamine transporter ASCT2 (Supp Fig. 3c), this metabolic profile exemplified a dramatic activation of the glutaminolysis pathway in 4-1BB CAR-Tregs.

Tyrosine phosphorylation signals, as assessed by western blot analysis, were markedly different among 4-1BB CAR-Tregs,

CD28 CAR-Tregs and untransduced Tregs (Fig. 2e). Tyrosine phosphorylation of several proteins increased, including proteins with molecular weights of 115–120 kDa and 55 kDa in the cell lysates of 4-1BB CAR-Tregs and CD28 CAR-Tregs, respectively, compared to those of untransduced Tregs. Further analyses of specific pathways revealed that the phosphorylation of ERK 1/2 was greater in 4-1BB CAR-Tregs than in CD28 CAR-Tregs and untransduced Tregs, indicating overactivation of the downstream MAP kinase pathway (Fig. 2f, g). Moreover, Akt phosphorylation on Ser473 was increased in CAR-Tregs compared to untransduced cells, irrespective of the CSD (Fig. 2f). Taken together, these findings demonstrate that CAR-Treg proliferation, induced by the 4-1BB-CAR tonic signal is associated with significant activation of the Akt/mTORC1, glutaminolysis and MAPK pathways.

**4-1BB tonic signaling does not result from greater CAR expression.** Ligand-independent CAR tonic signaling frequently depends on the high cell-surface density and/or intrinsic characteristics (scFv and hinge/spacer domains) of the CAR, which together entail CAR clustering[21]. Hence, we compared CAR expression according to the CSD and cell lineage. Although the vector copy number (VCN) in transduced Tregs was similar between CD28 CAR and 4-1BB CAR (Fig. 2h), 4-1BB CAR-Tregs displayed stronger binding to the HLA-A2 pentamer and greater EGFR expression than either CD28 CAR-Tregs or CAR-Tconvs (Supp Fig. 4a). We next examined whether this flow-cytometry pattern indicated greater vector expression in 4-1BB CAR-Tregs. Since the flow-cytometry pattern was tightly correlated with the blastic phenotype (Supp Fig. 4a), CAR and EGFR expression was normalized to the cell-surface area (Supp Fig. 4b). Normalized mean fluorescence intensities (RFIs) of HLA-A2 pentamer and protein L, a recombinant protein that binds to the kappa light chain without interfering with the antigen-binding site, still tended to indicate a greater expression of 4-1BB CAR than of CD28 CAR in Tregs (Supp Fig. 4b). In contrast, normalized EGFR expression was similar across CAR CSDs (Supp Fig. 4b), although stoichiometric co-expression of CAR and EGFRt was expected by the use of T2A. In fact, levels of CD28- and 4-1BB-CAR-T2A-EGFRt mRNA were strictly comparable, irrespective of the part (scFv vs. EGFRt) of the bicistronic mRNA amplified by RT-PCR (Fig. 2i).

From these observations, we inferred that 4-1BB tonic signaling, predominantly observed in Tregs, could not be ascribed to a greater transduction efficacy or to an increased expression of the mRNA-encoding 4-1BB CAR. Instead, these results might suggest that the tonic signal itself induces posttranslational CAR modification and/or trafficking impairment, accounting for increased CAR, but not EGFRt, expression at the cell membrane.

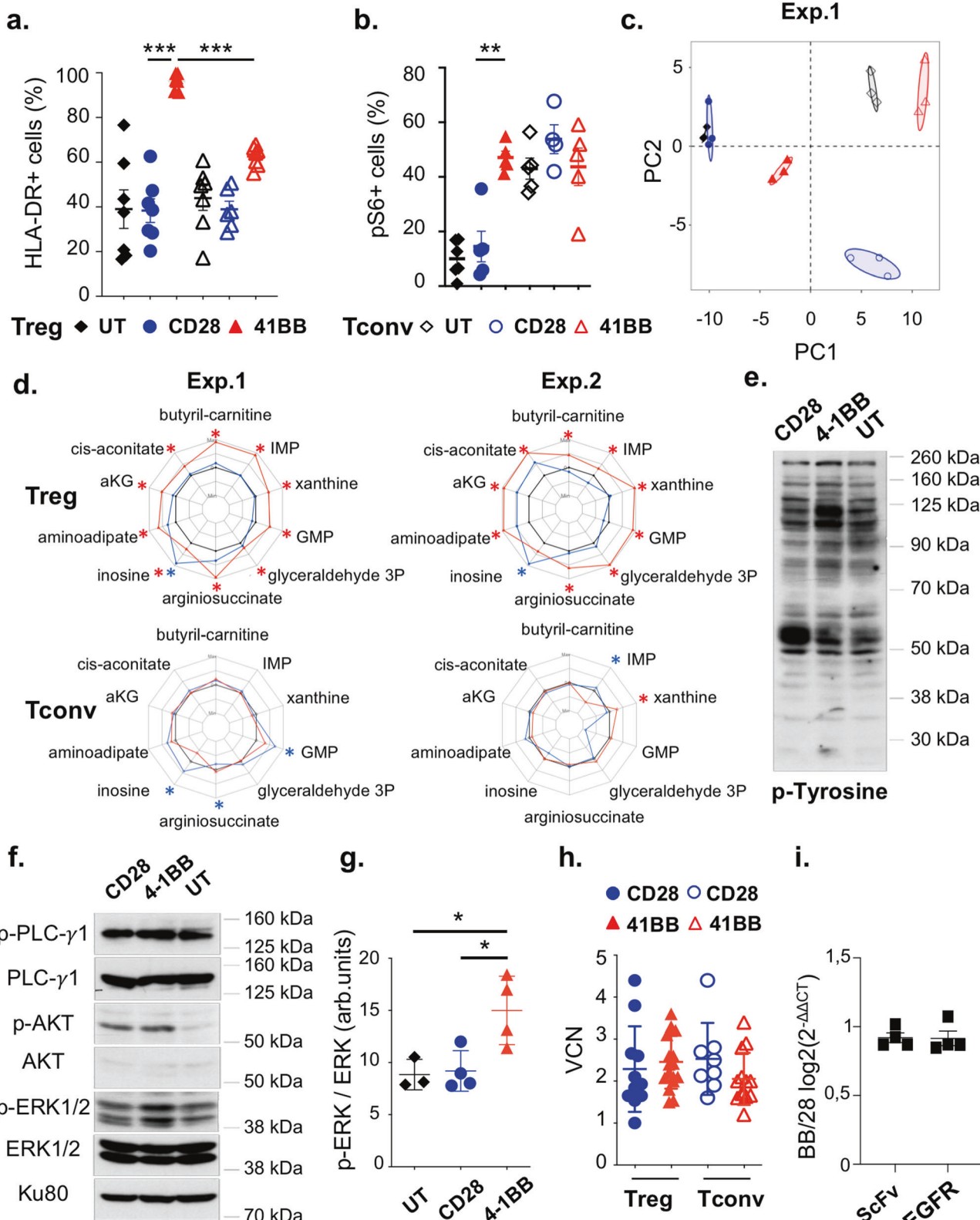

**4-1BB and CD28 tonic signals induce different Treg effector programs**. We next explored whether the differential effects of CD28 and 4-1BB tonic signaling on CAR-Treg proliferation would have similar impacts on the CAR-Treg phenotype. To compare the effects of the 4-1BB and CD28 CSDs on CAR-Tregs, we applied the t-distributed stochastic neighbor embedding algorithm (tSNE) to visualize high-dimensional cytometry data

on a 2-dimensional map at single-cell resolution[39]. Figure 3a shows the tSNE map for a pool of CD28 CAR-Treg and 4-1BB CAR-Tregs from three different donors. Strikingly, the CD28 CAR-Treg and 4-1BB CAR-Treg populations were clustered and highly separated from one another in a dichotomous manner, likely resulting from the sharp difference in HLA-DR expression. Of note, CD15s expression was limited to a small subset of

**Fig. 2 4-1BB tonic signaling induces strong CAR-Treg activation associated with the mTOR and MAPK pathways. a** HLA-DR-positive cell frequency, as assessed by flow cytometry at D16, is compared across CD28 and 4-1BB CAR-Treg and -Tconv populations with a two-tailed Mann–Whitney test. 4-1BB CAR-Tregs display a higher HLA-DR-expressing cell frequency than CD28 CAR-Tregs and 4-1BB CAR-Tconvs: ***$p = 0.0003$ (Mean $+/-$ SEM). $N = 7$ donors over seven independent experiments. **b** The frequency of phosphoS6-positive determined by flow cytometry (day 16). $N = 5$ donors from five independent experiments. Kruskal–Wallis test in Tregs only, **$p < 0.01$, $p = 0.0022$ (Mean $+/-$ SEM). **c** Principal component analysis (PCA) representing the UT, CD28, and 4-1BB CAR-Treg and corresponding Tconv groups based on the top 2 most significantly contributing components of metabolomics data. **d** The radar plots represent the log2-fold-change (log2(FC)), from minimal to maximal values, in metabolite levels, calculated as the ratio between CAR populations and their respective untransduced controls, from two independent experiments; adjusted $p$-values <0.05 are indicated with an asterisk (CD28 (blue), 4-1BB (red) and untransduced (black). **e** Tyrosine phosphorylation signal profile assessed by western blot, representative of three independent experiments (molecular weights shown on the right). **f** Phosphorylated PLC-γ1, phosphorylated ERK1/2, phosphorylated Akt, and Ku80, a loading control, are shown. Molecular weights are shown on the right. The immunoblot is representative of four independent experiments. **g** The phospho-ERK signal intensity relative to the ERK signal intensity is depicted in arbitrary units (arb.units) for four independent experiments. Mean $+/-$ SD are shown. 4-1BB CAR-Tregs display a higher ERK phosphorylation than UT (*$p = 0.0307$) and CD28 CAR-Tregs (*$p = 0.026$) when compared with two-tailed unpaired $t$-tests. **h** Vector copy number on day 16. Each replicate and the medians with interquartile range (IQR) are represented. At least $n = 10$ donors from 10 independent experiments are shown. **i** CAR mRNA expression measured by RT-qPCR using two sets of primers (scFv and EGFR sequences) on day 16. Ratios of expression of 4-1BB/CD28 CAR-Tregs (Mean $+/-$ SEM) are representative of four independent experiments. Color code: Blue full circle = CD28 CAR-Tregs, red full triangle = 4-1BB CAR-Tregs, black full diamond = UT Tregs, blue open circle = CD28 CAR-Tconvs, red open triangle = 4-1BB CAR-Tconvs, black open diamond = UT Tconvs.

HELIOS-negative CD28 CAR-Tregs. The CD28 CAR-Tregs maintained CCR7 expression overall, in contrast to their 4-1BB counterparts. Interestingly, a small subset of the 4-1BB CAR-Treg population expressed high levels of FOXP3 and HELIOS along with markers of activation (HLA-DR, 4-1BB, and ICOS) and suppressive functions (TIGIT and CTLA-4). This subset, which was not observed in the CD28 CAR-Tregs, met the phenotypic criteria of effector Tregs[40].

In order to gain further insights into the biological changes induced by CAR-tonic signal in Tregs, according to the CSD, we performed a bulk RNA-seq analysis in 4-1BB CAR- and CD28 CAR-Tregs. In keeping with flow-cytometric data, we found an increased expression of HLA class II and related (*CD74, CIITA*) genes in 4-1BB CAR-Tregs compared to CD28 CAR-Tregs (Fig. 3b). Importantly, the two CAR-Treg populations did not differ with regard to the expression of a well-established core Treg signature (Fig. 3c)[41]. However, the two populations displayed markedly different expression levels of transcripts, differentially expressed in activated vs. resting Tregs (Fig. 3d). 4-1BB and CD28 CAR-Tregs were significantly enriched in genes upregulated and downregulated in activated Tregs, respectively. This finding highlighted the greater state of tonic-signal-induced activation in 4-1BB CAR-Tregs than in CD28 CAR-Tregs (Fig. 3d). Notably, though, both CAR-Treg populations expressed genes related to Treg suppressive function, yet involving different molecular pathways. *ENTPD1* (CD39) and *EBI3* (IL35) had a greater expression in 4-1BB CAR-Tregs (Fig. 3c, e), whereas *IL10*, *LAG3*, and *HAVCR2* (TIM3) were significantly more expressed in CD28-CAR-Tregs (Fig. 3b, e).

**4-1BB tonic signaling precipitates CAR-Treg destabilization**. Tight control of the Akt/mTOR, MAPK and glutaminolysis pathways was found to be critical for the maintenance of Treg function[28,42,43]. Therefore, we analyzed the stability of the Treg lineage according to the CAR CSD. To this end, the expression of FOXP3 and HELIOS was assessed throughout the culture period. On day 16, the proportion of FOXP3+ (Fig. 4a, b) was similar across Treg populations and was much greater than that of Tconvs. However, a double-negative FOXP3- HELIOS- population was readily detected as early as day 16 in 4-1BB CAR-Tregs and subsequently expanded in all conditions (Fig. 4c). Furthermore, the loss of HELIOS expression among FOXP3+ cells was significantly precipitated in CD28 CAR-Tregs compared to other populations (Fig. 4a, d). Therefore, we examined the demethylation status of the Treg-specific demethylated region (TSDR) and

found that CAR-Tregs retained an overall high degree of TSDR demethylation on day 16, irrespective of the CSD (Fig. 4e). A recent study raised the concern that 4-1BB CARs could switch on a cytotoxic program in human Tregs[10]. In this respect, none of the CAR-Treg populations produced inflammatory cytokines (TNF-α or IFN-γ), in contrast to CAR-Tconvs. It is, however, worth noting that CD28 CAR-Tregs produced significantly more IL-10 than 4-1BB CAR-Tregs and untransduced Tregs, in line with transcriptomic data (Fig. 4f). Furthermore, CAR-Tregs did not induce HLA-A2-specific cytotoxicity over 10 h, regardless of the CSD, in contrast to CD8+ CAR-Tconvs (Fig. 4g and Supp 5), nor did they express granzyme B (Fig. 4h). Overall, these results did not support a drift in CAR-Tregs, regardless of the CSD, toward a cytotoxic/proinflammatory program.

We concluded that CAR-Treg manufacturing did not unduly disrupt Treg stability, or alter the noninflammatory profile of CAR-Tregs, regardless of the CSD. However, the emergence of a double-negative FOXP3- HELIOS- population raised the concern that prolonged culture and extensive proliferation, especially that driven by 4-1BB tonic signaling, may promote Treg instability.

**The combination of rapamycin and vitamin C abates 4-1BB tonic signal**. Uncontrolled mTOR pathway activation and hyperglutaminolytic states promote Treg destabilization[28,37,43]. Therefore, we hypothesized that mTOR inhibition mitigates metabolic reprogramming and prevents 4-1BB CAR-Treg instability[44]. Moreover, glutamine metabolites inhibit ten eleven translocation (TET) enzymes that demethylate the *FOXP3* gene[43], whereas vitamin C, a potent activator of TET enzymes, promotes Treg stability[45,46]. We next examined whether culturing 4-1BB CAR-Tregs with a Treg-supporting cocktail, including the mTOR inhibitor rapamycin and the epigenetic modifier vitamin C, would improve maintenance of the 4-1BB CAR-Treg lineage. Rapamycin and vitamin C were added to the culture medium at the time of restimulation (day 11) and compared to complete medium alone. As expected, the addition of rapamycin and vitamin C strongly inhibited mTORC1 activity (Fig. 5a). In line with these findings, the expansion rate of 4-1BB CAR-Tregs decreased to a level comparable to that of of CD28 CAR-Tregs (Fig. 5b). As a positive effect, reduced proliferation was correlated with greater 4-1BB CAR-Treg stability, according to phenotypic (Fig. 5c) and epigenetic assessments (Fig. 5d). For subsequent in vivo experiments, the Treg-supporting cocktail was added (pretreated) or not (untreated) to the culture medium for 4-1BB CAR-Tregs, following CD3/CD28 restimulation, at day 11.

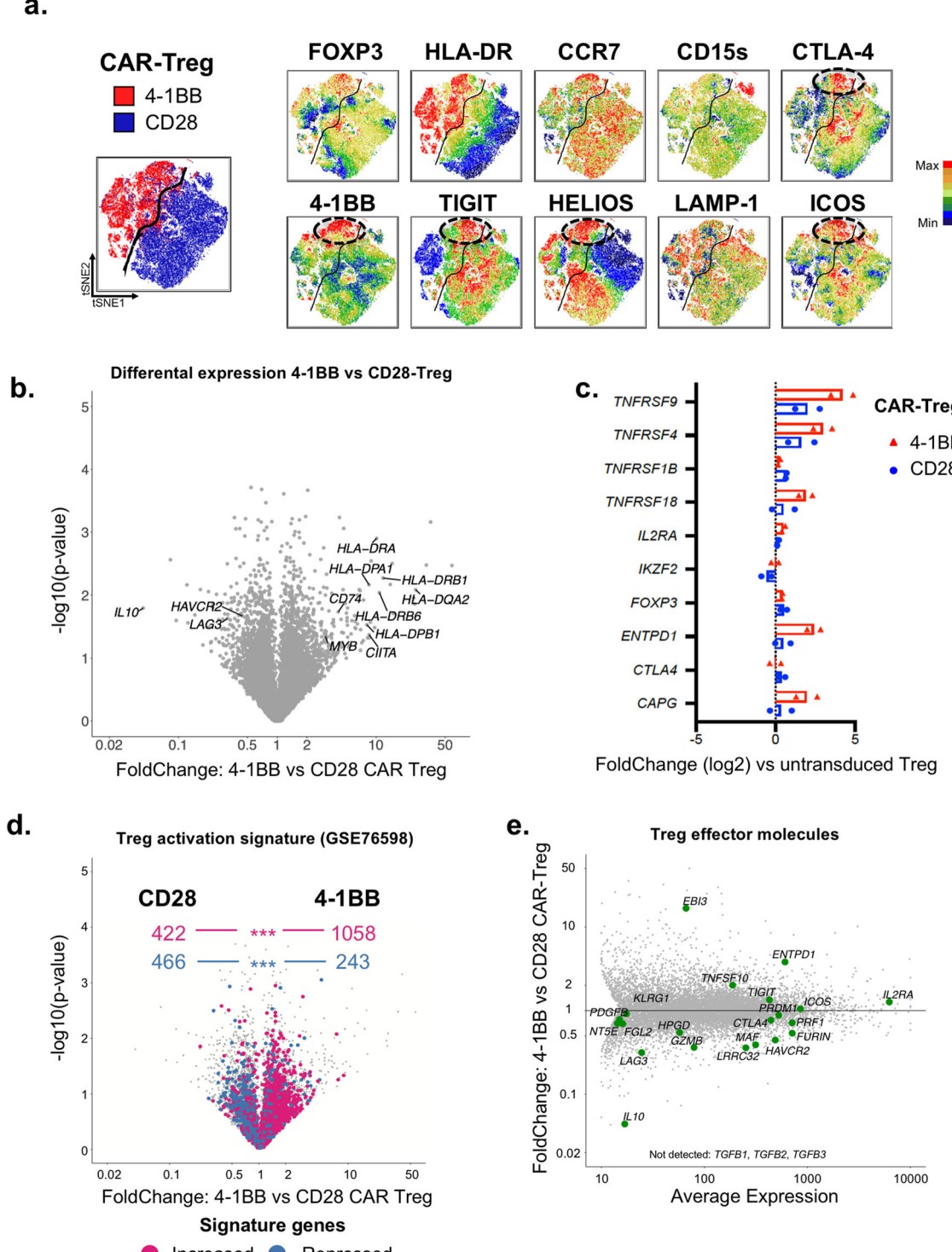

**Fig. 3 4-1BB and CD28 tonic signaling induces different Treg differentiation pathways. a** tSNE maps of CD28 and 4-1BB CAR-Tregs on day 16 are shown. Each point in the tSNE maps represents an individual cell, and cells are colored according to the intensity of fluorescence of the indicated markers. $n = 3$ donors pooled from three independent experiments. **b–e** All of the transcriptomic data come from two independent experiments in which living untransduced Tregs (UT), 4-1BB CAR-Tregs and CD28 CAR-Tregs were FACS-sorted on day 16 of culture. **b** Volcano plot (Fold-Change [FC] vs. − log 10 (p-value)) displaying the transcriptomes of 4-1BB CAR-Tregs vs. CD28 CAR-Tregs. **c** Bar graph showing the expression of the human core Treg genes from Zemmour et al., 2018[41] for each CAR-Treg population (blue bars = CD28, red bars = 4-1BB), in comparison with untransduced Tregs. **d** A Treg activation gene signature from[69] is mapped on the 4-1BB vs. CD28 volcano plot (same as **b**.). Values in the top half represent the number of genes, either upregulated (purple) or downregulated (blue) in the Treg activation signature, according to their enrichment in 4-1BB vs. CD28 CAR-Tregs. p-values for the signature enrichment were obtained by $\chi^2$ test. ***$p < 0.0001$. **e** MvA plot showing the average expression of the key genes involved in Treg effector function according to their differential expression between 4-1BB and CD28-CAR-Tregs.

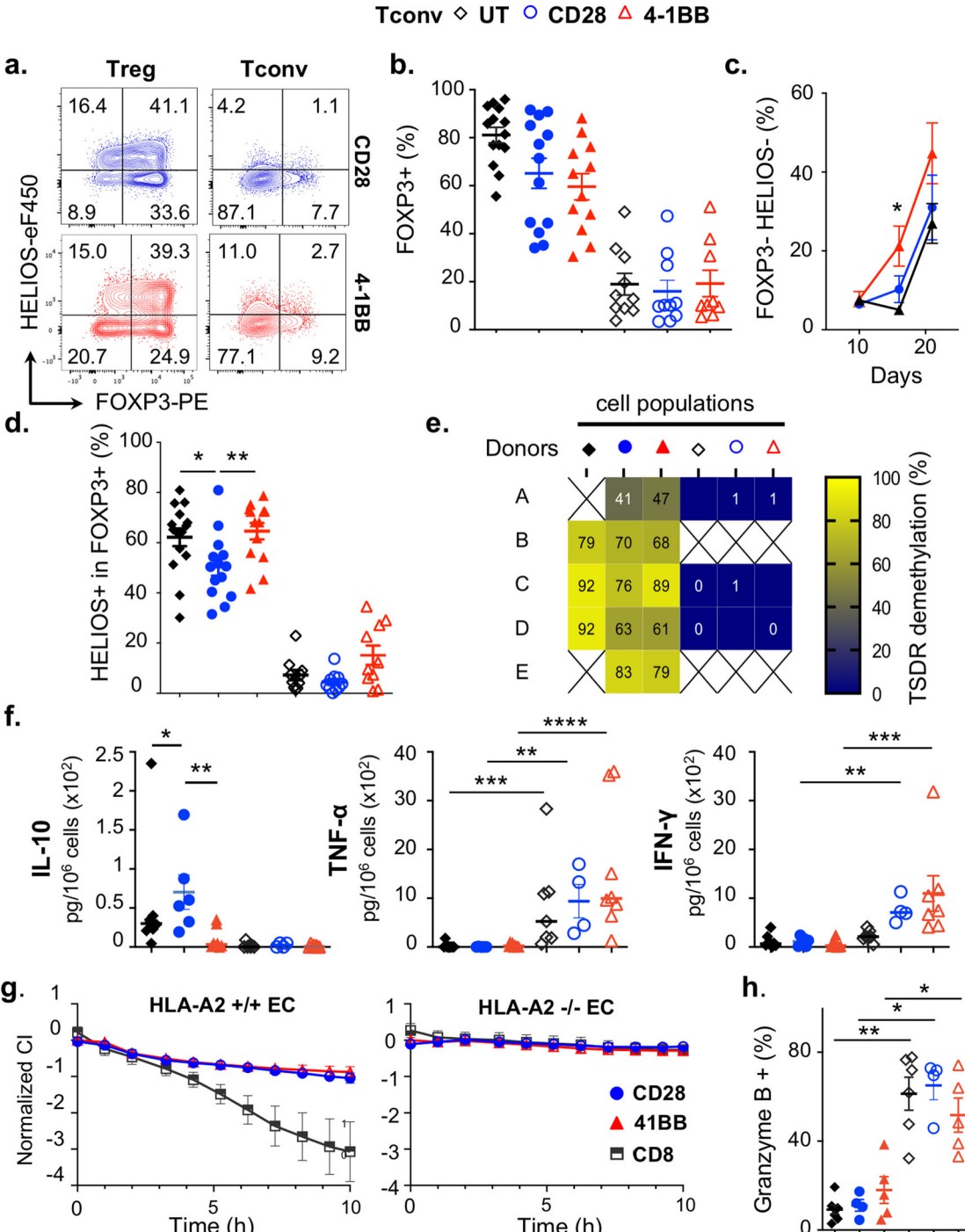

**Rapamycin/vitamin C cocktail rescues 4-1BB CAR-Treg function**. To assess the suppressive capacities of CAR-Tregs in vivo in an antigen-specific manner, we used a xenogeneic GVHD model based on the transfer of peripheral blood mononuclear cells (PBMCs) into busulfan-conditioned NOD SCID Gamma (NSG) mice. Different doses of hPBMCs were tested (5, 10 or 20 × 10[6] hPBMCs), demonstrating a dose-dependent effect on human cell engraftment, GVHD score and survival (Supp Fig. 6a–c). Of note, levels of human IFN-γ in the plasma were significantly correlated with human cell engraftment (Spearman r = 0.74; Supp Fig. 6d). The primary target organs included the lungs and liver, which were massively infiltrated by leukocytes (Supp Fig. 6e).

Next, we wanted to compare HLA-A2 CAR-Tregs to control polyclonal Tregs (transduced with an EGFRt-expressing vector, referred to as MOCK-Tregs) with respect to xenoGVHD prevention efficacy. Notably, 5 × 10[6] hPBMCs and 5 × 10[6] Tregs were injected at the same time through different intravenous routes (retroorbital sinus and tail vein). This precaution stemmed from a preliminary study showing a trend toward reduced CAR-Treg recirculation in the spleen when CAR-Tregs were mixed with HLA-A2+ PBMCs before infusion (not shown). CAR-Tregs and MOCK-Tregs were cotransduced with a lentivirus encoding luciferase and the mCherry reporter gene to allow in vivo tracking (Fig. 6a). Regarding survival, all of the mice transferred with

**Fig. 4 4-1BB tonic signal precipitates CAR-Treg destabilization. a–d** Flow-cytometry analysis of HELIOS and FOXP3 expression. **a** Representative contour plots on day 16. **b** Frequency of FOXP3-positive cells on day 16 from at least 10 independent experiments. **c** Frequency of double-negative (HELIOS and FOXP3) cells on days 10, 16, and 21 from 11 independent experiments. Two-tailed Wilcoxon matched-pairs signed rank test at day 16: $p = 0.0156$. **d** Frequency of HELIOS+ cells among FOXP3+ cells is lower in CD28 CAR-Tregs than in UT Tregs ($p = 0.0172$) and 4-1BB CAR-Tregs ($p = 0.0075$) on day 16 from 10 independent experiments. **e** qPCR analysis on day 16 showing the percent demethylation of the Treg-specific demethylated region (TSDR) from five independent experiments **f** On day 14, cytokines levels were measured by cytometric bead array analysis of culture supernatants and normalized to the number of cells. UT Tregs ($n = 8$), CD28 CAR-Tregs ($n = 6$), 4-1BB CAR-Tregs ($n = 9$), UT Tconvs ($n = 7$), CD28 CAR-Tconvs ($n = 4$) and 4-1BB CAR-Tconvs ($n = 8$), from nine independent experiments. Statistical significance was found in the following comparisons: IL-10, UT vs. CD28 CAR-Tregs (*$p = 0.0379$), 4-1BB vs. CD28 CAR-Tregs (**$p = 0.0047$); TNFα in Tconvs vs. Tregs, UT (***$p = 0.0006$), CD28 (**$p = 0.0095$), 4-1BB (****$p < 0.0001$); IFNγ in Tconvs vs. Tregs, CD28 (**$p = 0.0095$), 4-1BB (***$p = 0.0002$). **g** HLA-A2+ endothelial cells (Ecs) or HLA-A2- Ecs were cocultured with CD28 ($n = 4$ independent experiments) or 4-1BB ($n = 3$ independent experiments) CAR-Tregs or CD8+ Tconvs ($n = 4$ for HLA-A2+ and $n = 3$ for HLA-A2- cocultures). The normalized cell index is depicted. **h** Frequency of granzyme B-positive cells. UT Tregs ($n = 7$), CD28 CAR-Tregs ($n = 4$), 4-1BB CAR-Tregs ($n = 5$), UT Tconvs ($n = 6$), CD28 CAR-Tconvs ($n = 4$) and 4-1BB CAR-Tconvs ($n = 5$) from seven independent experiments. Statistical significance is observed between Tconv and Treg populations: UT ($p = 0.0012$), CD28 ($p = 0.0286$), 4-1BB ($p = 0.0159$). Black diamond, blue circle and red triangle were used to refer to UT, CD28 and 4-1 BB populations, respectively (full symbols = Tregs, open symbols = Tconvs), and black squares indicated CD8 4-1BB CAR-Tconvs. In panels **b**, **c**, **d**, **f**, **g**, and **h**, data are depicted as mean +/− SEM and compared using two-tailed Mann–Whitney test, unless otherwise indicated (*$p < 0.05$, **$p < 0.01$, ***$p < 0.001$, ****$p < 0.0001$).

PBMCs alone died within 60 days, as expected, whereas polyclonal MOCK-Tregs and untreated 4-1BB CAR-Tregs slightly delayed the disease course (median survival: 36 days for PBMCs, 43 days for MOCK-Tregs, $p = 0.016$, 52 days for 4-1BB CAR-Tregs, $p = 0.004$). In contrast, both pretreated HLA-A2-specific 4-1BB CAR-Tregs and HLA-A2-specific CD28 CAR-Tregs produced significant protection against xenoGVHD and related death (Fig. 6b). Moreover, circulating hIFN-γ levels were significantly lower at day 24 in mice that received either CD28 CAR-Tregs or pretreated 4-1BB CAR-Tregs than in animals transferred with PBMCs alone, mock-Tregs or untreated 4-1BB-CAR-Tregs (Fig. 6c). These results agreed with those for human chimerism in the blood, which was strongly inhibited by cotransfer of pretreated 4-1BB- and CD28-CAR-Tregs, whereas the cotransfer of untreated 4-1BB-CAR Tregs seemed to suprisingly boost early expansion of human PBMCs (Fig. 6d). Similarly, histological scores revealed better control of tissue inflammation, especially in the liver, in the group receiving pretreated 41BB-CAR-Tregs than in controls (Fig. 6e).

**4-1BB tonic signaling impairs in vivo homeostasy and function of CAR-Tregs.** Strikingly, circulating pretreated 4-1BB and CD28 CAR-Tregs were readily detected from day 10 to day 24 (Fig. 7a), far later than in previous reports[14,20]. In our hands, a sizeable population of CD28 CAR-Tregs was detected in the blood (median: 2.2 [0–9.1]% of circulating cells) as late as day 60–62, which was when mice were sacrificed (Supp Fig. 7a). In contrast, polyclonal MOCK-Tregs and untreated 4-1BB CAR-Tregs vanished very rapidly from the peripheral blood (Fig. 7a) and were barely detectable after day 10.

Luciferase-expressing CAR-Tregs were found to accumulate in the spleen (Fig. 7b). Using the In Vivo Imaging System (IVIS) Spectrum optical/CT coregistration system, we were able to not only map CAR-Tregs reliably in organs but also provide accurate quantification of cells in a 3D-specific region of interest, i.e., the spleen (Fig. 7b). In line with the blood data, we found that MOCK-Tregs rapidly disappeared, whereas pretreated 4-1BB and CD28 CAR-Tregs exhibited an expansion peak at day 12 of greater magnitude than untreated 4-1BB CAR-Tregs (Fig. 7b).

Furthermore, CD28 CAR-Tregs were detected in the spleen longer than pretreated 4-1BB CAR-Tregs throughout the follow-up period after day 12. Indeed, the number of splenic CD28 CAR-Tregs plateaued and remained constant from day 12 to day 58, whereas that of pretreated 4-1BB CAR-Tregs slowly decreased over time until reaching a significant decrease on day 58 (Fig. 7b).

We concluded that transient ex vivo mitigation of the 4-1BB tonic signal largely rescued the subsequent abilities of 4-1BB CAR-Tregs to expand, suppress and persist in vivo.

**4-1BB tonic signal reduces the ability of CAR-Tregs to be stimulated.** We next wondered whether the 4-1BB-tonic signal was associated with a defective CAR signal transduction, similar to that observed in effector T cells[21]. To address this issue, cultured CAR-Tregs and untransduced Tregs, which were collected after 16 days of culture, were stimulated overnight with either anti-CD3/CD28 beads or plate-bound HLA-A2 pentamers. Strikingly, 4-1BB CAR-Tregs failed to properly upregulate the expression of early activation markers, such as CD69 (Fig. 8a), especially after CAR stimulation. In sharp contrast, CD28 CAR-Tregs and polyclonal Tregs significantly induced CD69 expression in response to antigen receptor stimulation. Of note, pretreatment with an mTOR inhibitor did not restore CAR-induced CD69 expression nor did it restrain constitutive HLA-DR expression by 4-1BB CAR-Tregs. Importantly, 4-1BB CAR-Tregs were not completely hyporesponsive either, and they tended to increase the expression of 4-1BB and CD25 upon CAR stimulation (Fig. 8a).

4-1BB tonic signaling has only been reported following TCR-mediated activation[21]. The current paradigm thus proposes that an interaction between CAR 4-1BB CSD and TCR-associated adapter and/or signaling proteins accounts for an excessive activation of TNF receptor-associated factor (TRAF) downstream pathways. Therefore, we examined whether TCR deletion alleviates 4-1BB tonic signal and the ensuing effects. To this end, the *TRAC* gene was disrupted after a second round of anti-CD3/CD28 stimulation, using CRISPR-Cas9 gene editing (Fig. 8b). TCR-deficient 4-1BB CAR-Tregs were subsequently compared to TCR-positive 4-1BB CAR-Treg and polyclonal Treg controls. Following anti-CD3/CD28 bead removal, CAR-Tregs and polyclonal Tregs were rested for 3 days and then cocultured with either HLA-A2+ or HLA class I-deficient endothelial cell lines (Fig. 8b). First, TCR deletion, following TCR/CD3-mediated activation, did not change the blastic phenotype or HLA-DR expression, two hallmarks of 4-1BB tonic signaling in CAR-Tregs. Second, TCR-deficient 4-1BB CAR-Tregs still exhibited limited induction of CD69, but were able to upregulate 4-1BB and CD25 expression and to further increase cell size, upon CAR stimulation, in the same way as their TCR-positive counterparts.

Taken together, these findings suggest that 4-1BB tonic signaling impedes, but does not abrogate, CAR signal transduction following cognate antigen stimulation. Moreover, TCR

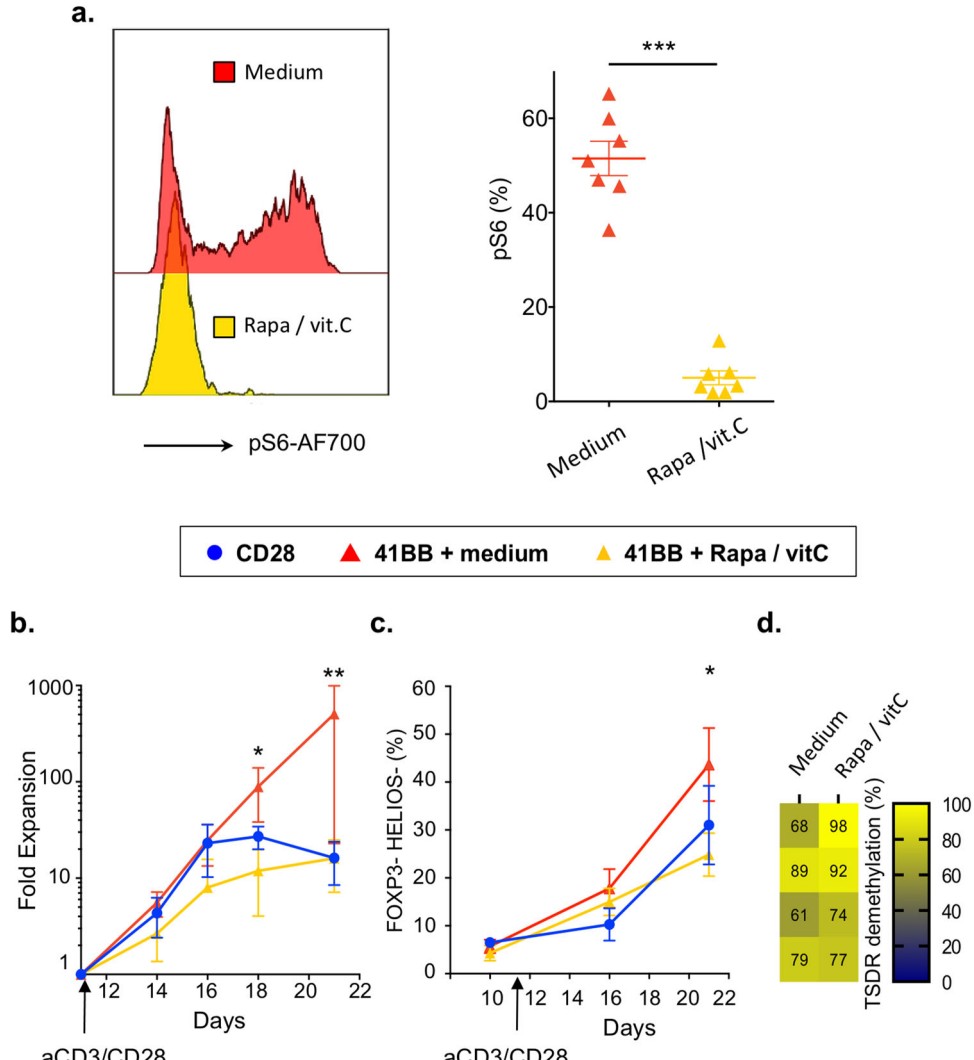

**Fig. 5 The combination of rapamycin and vitamin C abates overactivation due to 4-1BB tonic signaling and improves CAR-Treg stability. a** Histograms of phosphoS6 staining on day 16 of culture (left panel, representative of seven donors from seven independent experiments) and frequency comparison between 4-1BB and 4-1BB+ rapamycin/vitamin C ($p = 0.0006$) (right panel). **b** Fold expansion of CD28 or 4-1BB CAR-Tregs, cultured with or without rapamycin/vitamin C cocktail from the second anti-CD3/CD28 bead activation on day 11 (arrow) to day 21. $N = 5$ donors from five independent experiments. Statistical significance was found when comparing 4-1BB CAR-Tregs cultured with or without Rapamycin/vitamin C, at day 18 ($p = 0.0159$) and day 21 ($p = 0.0079$). **c** Frequency of double-negative (HELIOS and FOXP3) cells on days 10, 16, and 21 of culture by flow cytometry. Significant difference was found between 4-1BB CAR-Tregs cultured with ($n = 9$) or without ($n = 8$) Rapamycin/vitamin C cocktail, at day 21 ($p = 0.0499$). **d** Percentage of 4-1BB CAR-Tregs with Treg-specific demethylated region (TSDR) demethylation on day 16 of culture with or without rapamycin and vitamin C starting on day 11, from four different donors. Blue circle, red and yellows triangle refer to CD28, 4-1 BB, and 4-1BB + rapamycin/vitamin C populations, respectively (panels **a**–**c**). Data are depicted as mean +/− SEM (panels **a** and **c**) or median +/− interquartile range (IQR) (panel **b**) and compared using two-tailed Mann–Whitney test (*$p < 0.05$, **$p < 0.01$, ***$p < 0.001$, ****$p < 0.0001$).

disruption, following TCR-mediated activation, does not dampen 4-1BB tonic signaling.

## Discussion

Adoptive regulatory cell therapy represents a promising strategy for promoting operational tolerance in solid organ transplantation. Although substantial efforts have been made to understand the effects of CSD in CAR-T cells, this important issue has only recently begun to be investigated in Tregs[14,20]. In this study, we observed that both CAR-Treg subsets consistently exhibited specific phosphorylation patterns for downstream signaling proteins as well as unique transcriptomic and metabolic profiles, five days after TCR stimulation, compared to untransduced Tregs. These findings underscore the presence of tonic signaling

triggered by both CAR constructs with different ensuing biological effects.

In conventional T cells, the type of CSD incorporated in the CAR structure (CD28 vs. 4-1BB) was found to significantly influence CAR-T-cell metabolism, differentiation, survival and function, despite the recruitment of roughly the same signaling phosphoproteins, yet with various kinetics and strengths[47]. Briefly, upon activation, CD28/CD3 ζ CAR induces a more intense and faster signal than 4-1BB/CD3 ζ CAR[47], and promotes an aerobic glycolytic burst, which accompanies differentiation into effector memory T cells[48,49]. In contrast, the 4-1BB CSD promotes mitochondrial biogenesis and an oxidative metabolic program that supports long-lived central memory T cells[36,49,50]. Similarly, Dawson et al. found in Tregs that stimulation of CD28/CD3 ζ CAR, compared to their 4-1BB/CD3 ζ counterparts,

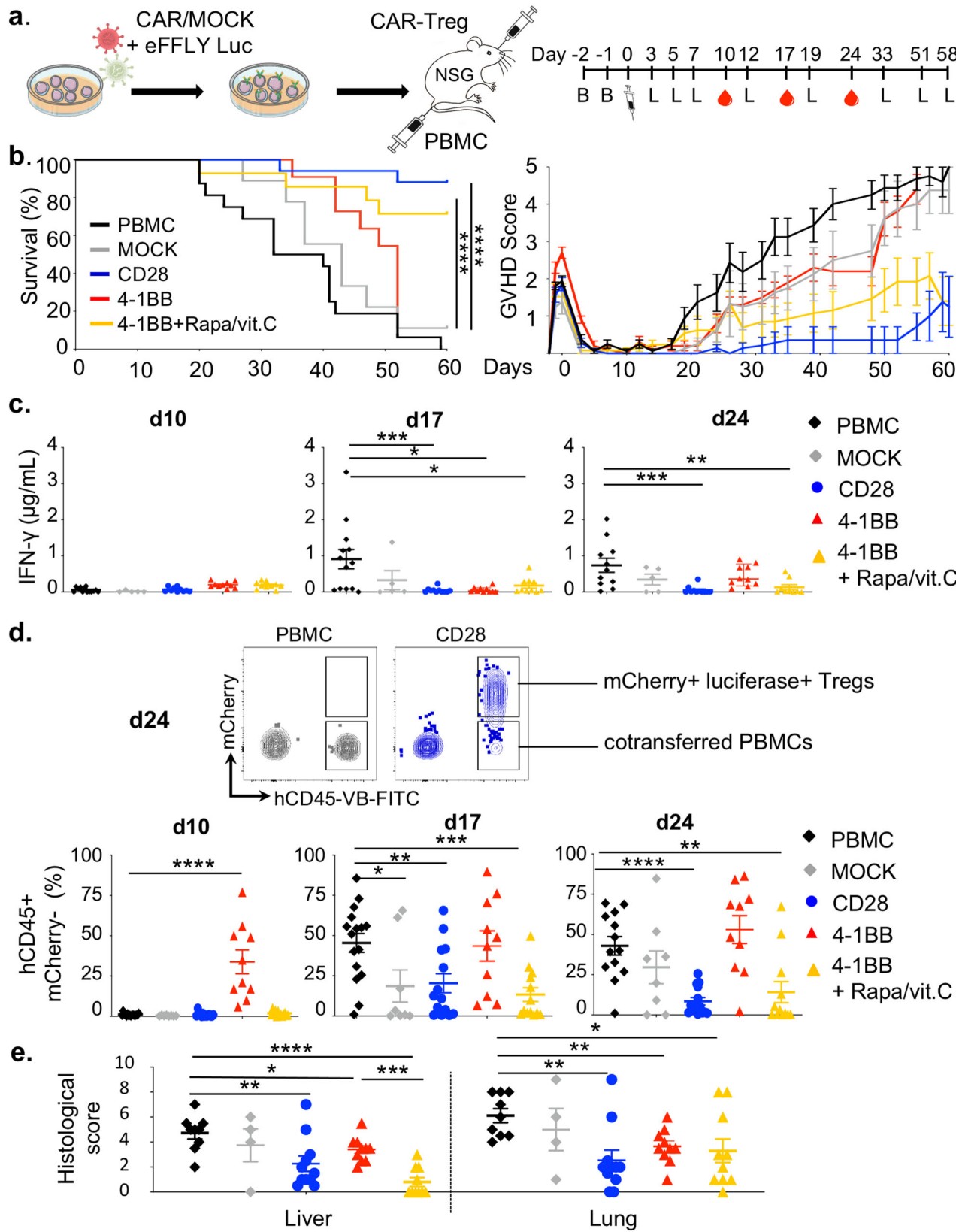

prompted enhanced glycolysis, involving both the mTORC1 pathway and Myc[14]. This process was associated with down-regulation of CCR7 and CD45RA, enhanced IL2/STAT5 signaling and increased expression of ICOS, CTLA-4 and MYB transcription factor[14], which are well-characterized hallmarks of effector Tregs[40]. In contrast, CCR7 expression was better preserved by 4-

1BB/CD3 ζ CAR-Tregs[14], in keeping with previous findings in Tconvs[25].

In regard to tonic signaling in CAR-Tconvs, the effect of CSDs becomes more muddled and at times conflicting due to the pleiotropic forms of tonic signaling inherent to the diversity of constructs and models[21]. However, when scFv clustering is

**Fig. 6 Rapamycin/vitamin C cocktail rescues 4-1BB CAR-Treg function.** Eight- to 12-week-old male NSG mice were conditioned with busulfan on days −2 and −1 before IV injection of $5.10^6$ HLA-A2+ PBMCs followed by retroorbital injection of $5 \times 10^6$ of Tregs (**b**–**d**): PBMC group ($n = 16$), CD28 CAR-Tregs ($n = 14$), 4-1BB CAR-Tregs ($n = 10$), 4-1BB + rapamycin/vitamin c ($n = 13$), and MOCK-Tregs ($n = 8$) from 4, 4, 1, 3, and 2 independent experiments, respectively. **a** Schematic design of the experiment. Mice were weighed and scored for GVHD three times a week and bled weekly for flow-cytometry analysis. **b** Survival curves (left panel) and GVHD scores (right panel). Log-rank Mantel-Cox tests comparing the indicated groups. Both CD28 and 4-1BB CAR-Tregs + rapamycin/vitamin c groups demonstrate a significantly better survival than the PBMC group (****$p < 0.0001$). **c** Plasma human IFN-γ levels were measured by cytometry bead array at days 10, 17, and 24. IFN-γ levels are significantly lower in CAR-Treg groups than in the PBMC group at day 17 (CD28, ***$p = 0,0004$; 4-1BB, *$p = 0,0147$, 4-1BB + rapamycin/vitamin C, *$p = 0,0147$) and at day 24 (CD28, ***$p = 0,0002$ and 4-1BB + rapamycin/vitamin C, **$p = 0,0073$). **d** Representative contour plots of circulating murine and human cells and frequencies of transferred PBMCs (lower panels). Compared to PBMC group, chimerism is statistically different at days 10 (4-1BB, ****$p < 0,0001$), 17 (MOCK, *$p = 0,0274$; CD28, **$p = 0,0060$; 4-1BB + rapamycin/vitamin C, ***$p = 0,0005$) and 24 (CD28, ****$p < 0,0001$; 4-1BB + rapamycin/vitamin C, **$p = 0,0030$). **e** Histologic analysis of liver and lung lesions at sacrifice from three independent experiments and three different donors, except for MOCK-Tregs (two experiments) and 4-1BB CAR-Tregs (one experiment). PBMCs only ($n = 9$), CD28 CAR-Tregs ($n = 11$), 4-1BB CAR-Tregs ($n = 10$), 4-1BB CAR-Tregs + rapamycin/ vitamin C ($n = 10$), and MOCK-Tregs ($n = 4$). Statistical significance is observed in the following comparisons: PBMC vs. 4-1BB + rapamycin/vitamin C, $p < 0.0001$ (liver) and $p = 0.0346$ (lung); PBMC vs. CD28, $p = 0,0076$ (liver) and $p = 0,0043$ (lung); PBMC vs. 4-1BB, $p = 0,04$ (liver) and $p = 0,0031$ (lung); and 4-1BB cultured with or without rapamycin/vitamin C $p = 0,0002$ (liver). Data are depicted as mean $+/-$ SEM (panels **b** to **e**) and compared using two-tailed Mann–Whitney test (**c** to **e**) (*$p < 0.05$, **$p < 0.01$, ***$p < 0.001$, ****$p < 0.0001$). Abbreviations: B, busulfan; L, bioluminescence.

involved, CD28 CSD is usually associated with a more deleterious effect than 4-1BB[21], regardless of whether the CAR tonic signal induces hastened terminal differentiation[22,51] or T-cell exhaustion[25]. 4-1BB CAR-T cells are not devoid of ligand-independent tonic signals either and two main phenotypes have been described. A positive feedback loop between 4-1BB-induced NFκB activation and the long terminal repeat (LTR) promoter of a nonself-inactivating retroviral vector can drive a tremendous induction of CAR transgene expression and subsequently cell death[23]. The use of a self-inactivated lentiviral vector, as in our study, protects against such a dramatic effect. In fact, we found that the expression of CAR mRNA did not differ between the CD28 and 4-1BB constructs. Alternatively, 4-1BB tonic signaling may support heightened in vitro expansion and a more prolonged blastic phase, which importantly does not curtail in vivo anti-tumor efficacy or CAR-T-cell survival[21,52]. A recent report described a transcriptomic signature of the 4-1BB CSD tonic signal, different from that of CD3ζ, associated with active cell division[36]. Notably, this study identified the high and sustained expression of HLA class II as a hallmark feature of 4-1BB CSD tonic signaling in Tconv[36]. We confirmed and extented this finding in Tregs. In addition, our study indicates that the 4-1BB tonic signal, induced by the same CAR construct, may boost cell proliferation, with associated metabolic changes, far more in Tregs than in Tconvs. Furthermore, 4-1BB-induced unleashed proliferation had a detrimental impact on Treg lineage identity and survival, whereas the 4-1BB tonic signal did not appear to compromise the in vivo efficacy of CAR-T cells[52].

The transcription factor MYB has been characterized as a key driver of the effector Treg program[40], and a hallmark of the most potent CAR-Tregs in vivo[14]. In this respect, 4-1BB CAR tonic signaling may result in strong baseline MYB expression (Fig. 3b)[53], a finding consistent with a sizeable population of FOXP3$^{High}$, HELIOS$^{High}$, CTLA-4$^{High}$, ICOS$^{High}$, and TIGIT$^{High}$ effector Tregs in 4-1BB CAR-Tregs at day 16 of culture[40].

There is growing evidence establishing the key role of metabolism in regulating immune cell fate decisions in a lineage-specific manner[30,54]. Human Treg proliferation requires both glycolysis and fatty acid oxidation, whereas the Tconv anabolic program relies on a switch from oxidative phosphorylation to aerobic glycolysis[30]. In Tregs, although mTORC1 activity is required for Treg proliferation and acquisition of full suppressive function[38,55], it is also well-established that the PI3K/Akt/mTOR pathway must be tightly regulated through a feedback loop involving phosphatase and tensin homolog (PTEN) to enforce Treg stability[28,56]. In fact, overactivation of mTOR[28] and related

anabolic programs, including glycolysis[37] and glutaminolysis[43], can precipitate Treg lineage instability. Hence, mTORC1 is increasingly recognized as the fulcrum of Treg regulation, balancing proliferative and suppressive capacities through an oscillatory switch[37,56]. In this respect, we found that 4-1BB tonic signaling was associated with dysregulated activation of the Akt/mTOR pathway and a strong induction of glutaminolysis, known to interfere with *FOXP3* epigenetics[43]. In fact, these changes were accompanied by a higher rate of HELIOS and FOXP3 loss in 4-1BB CAR-Tregs than in CD28 CAR-Tregs upon prolonged culture. Tonic signal-induced sustained Akt/mTOR activation in 4-1BB CAR-Tregs might also account for the rapid loss of CCR7 through forkhead box O (FOXO) 1 sequestration[57]. This finding seems to be in conflict with previous studies that emphasized better preservation of the central memory phenotype with 4-1BB than with CD28 CSD in both Tconvs[49] and Tregs[14]. We propose that this apparent discrepancy results from different timing and duration of Akt/mTOR activation between transient and constitutive CAR activation.

This study demonstrates that the tonic signal of 4-1BB significantly compromises the in vivo suppression and survival of CAR-Tregs. We believe that Treg lineage-specific effects reflects the critical importance of tight regulation of Akt/mTOR pathways in Tregs[28,56]. Similarly, DUSP4-dependent control of the MAPK pathway was found to be critical for the maintenance of Treg function[42]. Epigenetic marks along with constitutive expression of dual-specificity phosphatase 4 (DUSP4), a potent regulator of the MAPK superfamily, are highly conserved features of Tregs across species[19,58]. Together, these results suggest that dysregulated activation of MAPK and Akt/mTOR through constitutive 4-1BB signaling in Tregs could destabilize the Treg lineage and impede Treg function. These results are reminiscent of previous reports, in which 4-1BB CAR-Tregs lacked suppressive function or even switched on a cytotoxic program[10,14]. However, we found that 4-1BB CAR-Tregs, generated from highly purified naive Tregs, maintained the overall features of Tregs, including limited inflammatory cytokine production, reduced cytotoxicity, and TSDR demethylation. More importantly, the addition of a Treg-supporting cocktail to cultured 4-1BB CAR-Tregs reduced FOXP3 loss, and subsequently improved the in vivo suppressive function. We propose that transient mTOR inhibition recapitulates an enforced rest, able to restore CAR-Treg functionality, as recently shown in exhausted CAR-T cells with multikinase inhibition[59].

It was also proposed that the close proximity of 4-1BB CSD to the cell membrane in secondary CAR, but not in endogenous

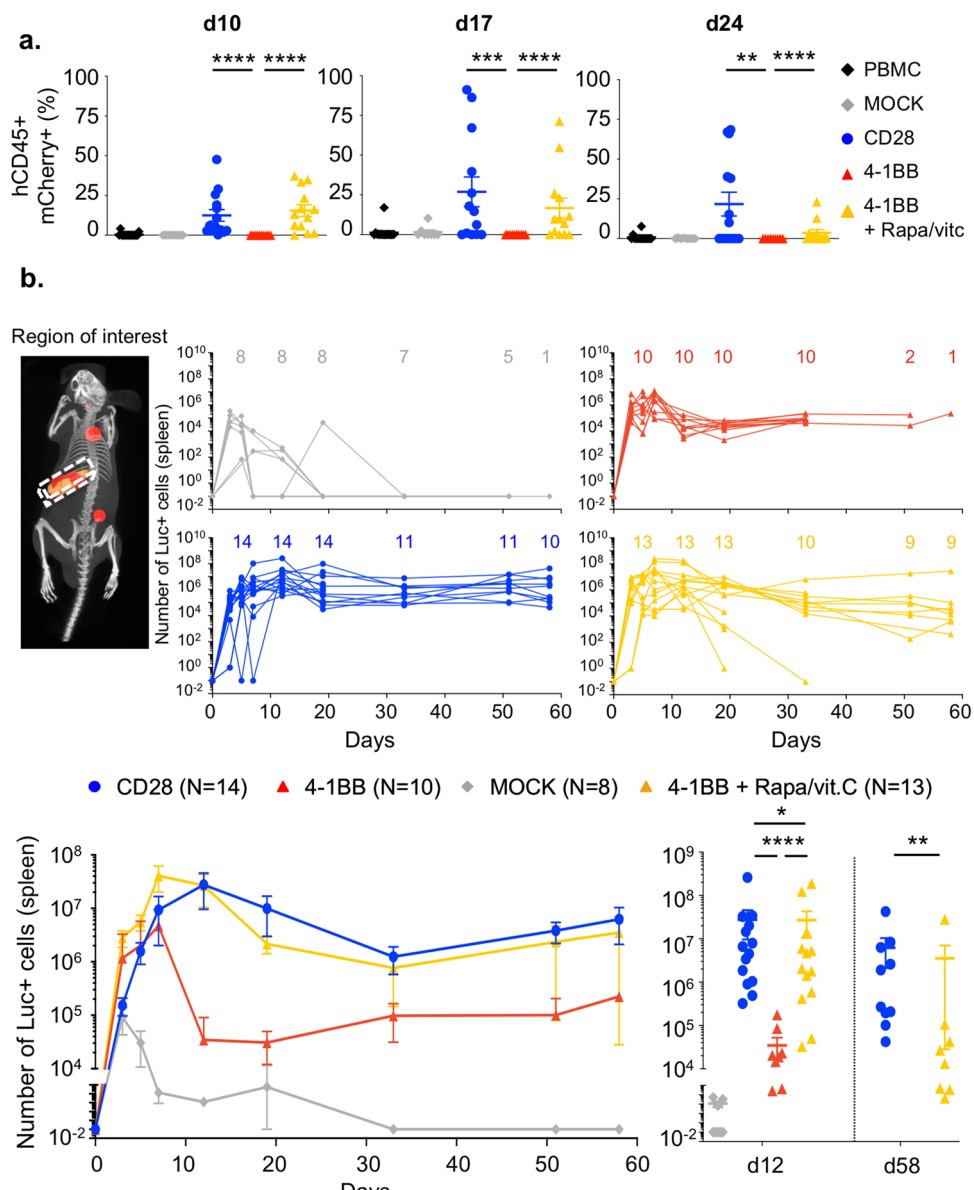

**Fig. 7 4-1BB tonic signaling limits the in vivo expansion and persistence of CAR-Tregs.** Panels **a** and **b** show results over three independent experiments and from three different donors, except for the groups MOCK-Tregs (two experiments) and 4-1BB CAR-Tregs (one experiment). **a** Frequencies of CAR-expressing (mCherry+) cells among the total circulating cells assessed by flow cytometry. Statistical significance is observed in the following comparisons: CD28 ($n = 14$) vs. 4-1BB ($n = 10$), $p < 0,0001$ (day 10), $p = 0,0010$ (day 17), $p = 0,0044$ (day 24); and 4-1BB vs. 4-1BB + rapamycin/vitamin C ($n = 13$), $p < 0,0001$ (day 10), $p < 0,0001$ (day 17), $p < 0,0001$ (day 24). PBMCs only $n = 16$, MOCK-Tregs $n = 8$. **b** 3D bioluminescence imaging tomography (DLIT) of recipient mice was performed as described in the Methods. A region of interest (ROI) corresponding to the spleen was determined by coregistration with the Automatic Mouse Atlas (upper left panel), and the absolute number of luciferase-positive (Luc+) cells was calculated. Individual kinetics of the absolute number of Luc+ cells for each mouse corresponding to each group and the number of living animals per timepoint are shown (upper panels). Grouped kinetics are shown in lower left panel. The number of mCherry-positive cells was measured in each group at day 12 (CD28 CAR-Tregs, $n = 14$; 4-1BB CAR-Tregs, $n = 10$; 4-1BB CAR-Tregs + rapamycin/vitamin C, $n = 13$; and MOCK-Tregs, $n = 8$ living animals) and day 58 (CD28 CAR-Tregs, $n = 10$; 4-1BB CAR-Tregs + rapamycin/vitamin C, $n = 8$ living animals) (lower right panel). Statistical significance is observed in the following comparisons: CD28 vs. 4-1BB, $p < 0.0001$ (day 12); MOCK vs. 4-1BB, $p = 0.0238$ (day 12); 4-1BB vs. 4-1BB + Rapa/vitC, $p < 0.0001$ (day 12); and CD28 vs. 4-1BB + Rapamycin/vitamin C, $p = 0.0085$ (day 58). Data are depicted as mean +/− SEM and compared using two-tailed Mann–Whitney test (*$p < 0.05$, **$p < 0.01$, ***$p < 0.001$, ****$p < 0.0001$).

4-1BB, facilitates the interactions between 4-1BB CSD and TRAFs, which in turn overly activate the NFκB pathway[21]. Consistent with this, 4-1BB tonic signaling was alleviated in CAR-Tconvs by spacing out the distance between the 4-1BB CSD and the cell membrane, through the incorporation of CD28 CSD in a 3rd generation CAR[21].

These data highlight the importance of systematic assessment of tonic signals in the process of CAR-Treg development. For instance, our results may shed light on the mechanisms underpinning the dysfunction of highly proliferative TNFR2 CAR-Tregs, as reported by Dawson et al.[14] TNFR2 CAR-Tregs demonstrated greater baseline expression of CD71 and LAP,

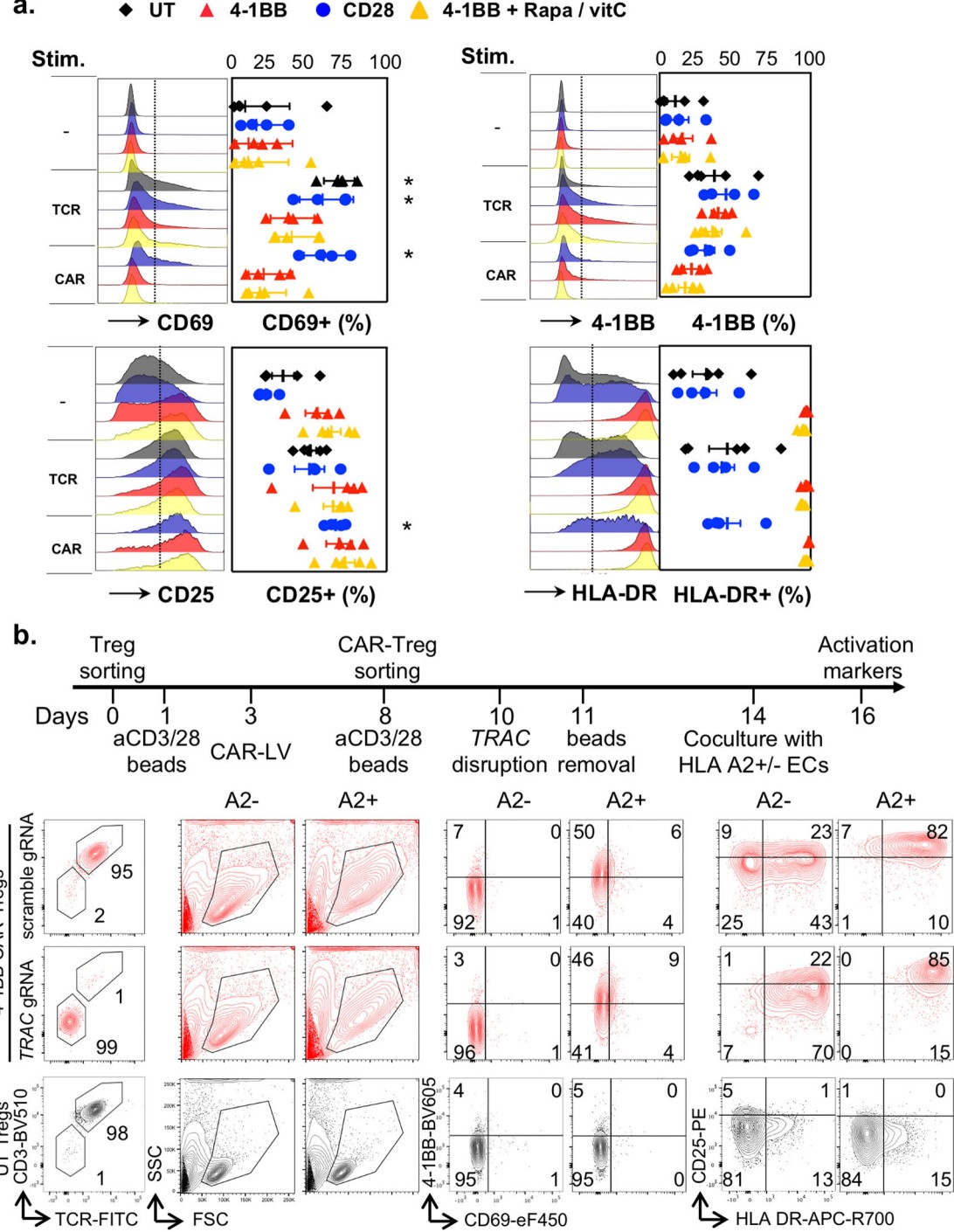

**Fig. 8 4-1BB tonic signaling reduces the ability of CAR-Tregs to be stimulated. a** Tregs were transduced and expanded as shown in Fig. 1c. On day 16, cells were stimulated overnight with either plate-bound HLA-A2 pentamers (CAR) or anti-CD3/CD28 beads (TCR) before flow-cytometry analysis. Representative histograms of each activation marker (left panels) and the frequencies of positive cells (right panels) before and after stimulation are shown. $N = 5$ donors from five independent experiments. The mean $+/-$ SEM is represented. Two-tailed Mann–Whitney tests were performed to compare unstimulated and stimulated (either via the TCR/CD3 complex or CAR) cells transduced with the same CAR construct. *$p < 0.05$. Statistically significant induction of CD69 marker is found in CD3/CD28-stimulated unstransduced (UT) Tregs ($p = 0.0159$), CD3/CD28-stimulated ($p = 0.0286$), and CAR-stimulated ($p = 0.0286$) CD28 CAR-Tregs. Similarly, CAR stimulation significantly increases the expression of CD25 by CD28 CAR-Tregs ($p = 0.0286$). **b** TCR-deficient or competent 4-1BB CAR-Tregs were produced as shown in the upper panel, *TRAC* was deleted using CRISPR/cas9 on day 10 of culture before being subsequently activated with HLA-A2 +/+ or HLA-A2−/− endothelial cells on day 14 and assessed for activation marker expression by flow cytometry on day 16. Contour plots representative of two independent experiments ($n = 2$ donors) show the expression of the indicated markers for *TRAC*-deleted 4-1BB CAR-Tregs, scrambled gRNA-negative control 4-1BB CAR-Tregs and UT Tregs (lower panel).

along with increased inflammatory cytokines upon CAR-independent stimulation, than CD28 CAR-Tregs[14].

Although, there are currently no universal guidelines for tonic signaling testing, it can be identified during primary Treg cell culture upon activation through CD3 and CD28 stimulation due to differences in phenotype and proliferation rates between CAR-Tregs and controls. Notably, increased tonic signaling in CAR Tconvs most frequently leads to increased expression of co-inhibitory markers, a terminally-differentiated phenotype, and a reduced antitumor effect. However, these surrogate markers may be of little value for Treg assessment. Moreover, our study in CD28 CAR-Tregs demonstrated that tonic signaling can result in slight changes, such as a hastened loss of HELIOS expression or increased IL-10 secretion. Therefore, we suggest thoroughly investigating Treg lineage stability, as well as in vivo survival and suppressive function. Furthermore, HLA-DR expression appears to be a useful indicator of constitutive 4-1BB signaling.

This study has one primary limitation. With respect to the impact of CAR tonic signaling in Tregs, we cannot affirm that the more detrimental effect of 4-1BB, compared to CD28 CSD, would also apply to other forms of CARs, especially those with self-aggregating properties, given the well-established protective effect of 4-1BB CSD in these settings[21,25]. Regarding Tregs, a preprint from Vancouver's group reports on the deleterious impact of tonic signaling of CD28 CAR in Tregs, using a self-clustering GD2-targeted CAR, compared to a nonself-aggregating CD19-targeted CAR[25,60]. GD2-specific CAR-Tregs exhibited higher expression of co-inhibitory molecules than CD19-specific CAR-Tregs devoid of tonic signals[60], similar to as their Tconv counterparts[25]. Importantly, GD2-targeted CAR-Tregs, with tonic signaling, failed to suppress in vivo despite Treg lineage stability[60]. It would be of interest to investigate whether 4-1BB CSD alleviates the negative effect of CAR clustering-related tonic signals, as shown in Tconvs[25]. Together, our study, along with the one reported by Lamarche et al. indicates the pleiotropic presentation of constitutive CAR activation in Tregs and paves the way for future investigations.

Finally, we were struck by the much longer persistence of circulating and lymphoid organ-resident CAR-Tregs in our xenoGVHD model than that reported in previous studies[14,61]. A simple explanation for these differences could be that we transferred CAR-Tregs and HLA-A2+ PBMCs through different routes (retroorbital sinus and tail vein) to avoid cell clustering and early trapping in the lungs following in vivo transfer. In fact, immediate encounter of CD19+ target B cells in the blood by CD19-specific CAR-T cells was found to trap the CAR-T cells in the lungs and reduce their access to lymphoid organs[62,63].

In conclusion, we found that constitutive CAR signaling can induce an imbalance between proliferation and suppressive function, whose equilibrium in Tregs is normally controlled by finely tuned Akt/mTOR and MAPK pathways and metabolic checkpoints. Hence, CAR tonic signaling interferes with the fate of CAR-engineered Tregs, similar to the effects on Tconvs. Notably, Tregs were highly sensitive to 4-1BB tonic signaling that precipitated Treg instability and in vivo dysfunction. However, transient ex vivo mitigation of 4-1BB tonic signaling dramatically improves CAR-Treg stability, expansion, function and persistence after transfer.

## Methods

**scFv specificity**. The specificity of different scFvs for HLA-A2-bound beads was assessed using a Luminex multiplex single antigen bead assay (LABScreen® Single Antigen). A cell-based assay was used to confirm HLA allele specificity. Tagged scFvs with low- and high-binding strength were incubated at three different concentrations, 0.05, 0.5, and 5 µg/mL, with PBMCs from HLA-A2+ and HLA-A2−/A28-donors. The mean fluorescence intensities (MFIs) of the tagged scFvs bound to

HLA-coated beads were measured by fluorescent detection using a laser-based instrument (Luminex™).

**CAR construct design**. The selected scFv was fused to a stalk region from the human CD8α hinge, the transmembrane domain from human CD8, the CD28 or 4-1BB CSD, and the human CD3ζ signaling domain. EGFRt was cloned downstream of T2A at the second gene position to serve as a reporter (Supp Fig. 1c).

**Cell source and purification of human Tregs**. Healthy donor PBMCs were obtained from the Etablissement Français du Sang. PBMCs were typed based on the expression or absence of HLA-A2/A28 molecules, as assessed by anti-HLA-A2/A28 antibody (OneLambda) staining evaluated by flow cytometry using the BD LSRFortessa™ X-20 analyzer. CD4+ T cells were enriched from HLA-A2- donor PBMCs using an EasySep CD4+ Enrichment kit (Stem Cell®). Naive regulatory T cells (nTregs), which were defined as CD4+ CD25Hi CD127− CD45RA+ CD45RO−, were sorted using a FACS Aria II (BD Biosciences). In parallel, we sorted CD4+ CD45RA+ CD25− Tconv cells from the same donor as controls. The sorting purity was checked with flow cytometry and always demonstrated a purity greater than 98%. Sorted T cells were stimulated with Dynabeads™ Human T-Activator CD3/CD28 (Thermo Fisher Scientific) (ratio 1:1) in X-VIVO® 20 medium containing 10% human serum AB (Biowest) and 1000 UI/mL human IL-2 (Proleukin, Novartis). HLA-typed CD3-depleted splenocytes were isolated from spleens collected from deceased organ donors through collaboration with the Regional Histocompatibility Laboratory and National Biomedicine Agency.

**T-cell transduction and expansion**. Two days after activation, Tregs were transduced with the different CAR constructs at a multiplicity of infection (MOI) of 30 viruses per cell. Prostaglandin E2 (PGE2) (Cayman Chemical, 10 µM) and a transduction adjuvant (Lentiboost, Sirion Biotech, 0.25 mg/mL) were added for a 6 h of transduction time. On day 7 posttransduction, CD4+ EGFRt+ cells were sorted using a FACS Aria II and then restimulated with anti-CD3/CD28 beads. T cells were expanded and cultured in complete medium supplemented with IL-2 (1000 UI/mL) for 7 to 10 days for in vitro experiments. Three Treg cultures (separate experiments) were interrupted because the FOXP3+ frequency among expanded untransduced Tregs had dropped below 50% at Day 10. These experiments were considered as failures, regardless the cause (sorting, donor, IL-2, …), and and excluded from further analysis.

**Flow cytometry**. Surface staining was performed using an HLA-A2 pentamer (Proimmune) and the monoclonal antibodies (clone, manufacturer, catalog number, and dilution) are listed in Supplemental Table 2. For CAR surface expression assessment, biotinylated protein L (Thermo Fisher Scientific) was used and bound to fluorochrome-conjugated streptavidin (BD Biosciences). For ASCT2 expression evaluation, a receptor-binding domain (Metafora) was preincubated for 20 min at 37 °C with cells, and an anti-mouse secondary antibody was used. pS6 staining was performed using a PerFix EXPOSE kit (Beckman Coulter). For evaluation of intracellular targets, cells were stained using a fixable viability dye (FVD, Thermo Fisher Scientific) before fixation, permeabilization and staining using a FOXP3/transcription factor staining buffer set (Thermo Fisher Scientific) according to the manufacturer's instructions. Intracellular staining was performed for FOXP3, HELIOS, Granzyme B and Ki67 (Thermo Fisher Scientific). Cells were analyzed using a BD LSRFortessa™ X-20 analyzer (BD Biosciences). Data were analyzed using Kaluza 2.1 (Beckman Coulter) or FlowJo 10.6.1 (BD Biosciences). For t-SNE mapping, the R Console plugin (https://www.beckman.fr/flow-cytometry/software/kaluza/r-console) was used to analyze flow-cytometry data in R within the Kaluza Analysis framework.

**Specific CAR-mediated activation in Jurkat RT3-T3.5 cells**. Jurkat J.RT3-T3.5 cells (ATCC; TIB-153) transduced with either a CD28 CAR vector or a 4-1BB CAR vector were cocultured overnight with irradiated (35 Gy) HLA-A2/A28-positive or HLA-A2/A28-negative human splenocytes at a cell ratio of 1:1. After 20 h of incubation, the cells were stained with an HLA-A2 MHC pentamer and anti-human EGFR antibody to monitor CAR expression; 7-aminoactinomycin D (7AAD) and an anti-human CD69 antibody were all used. Activation was defined as the upregulation of CD69 expression. Cross-reactivity was assessed against 8 different donors and up to 30 different allogeneic class I-HLA molecules.

**Activation assay**. Ninety-six-well flat-bottom plates were coated overnight at 4 °C or for 3 h at 37 °C with an HLA-A2 pentamer at 5 µg/mL in PBS. The plates were washed with cold PBS the next day or 3 h later. Cells were separated from anti-CD3/CD28 beads 24 h prior to seeding at 100,000 cells/well.

For the activation assay after genome editing, untransduced Treg, scramble and TRAC-edited CAR-Tregs were seeded with either HLA-A2+ or HLA-A2- endothelial cell lines at the indicated culture time. Activation markers were analyzed by flow cytometry after 24 and 48 h of coculture.

**Immunoblotting**. T cells (3–5 × 10^6) were lysed in 1% NP-40, 50 mM Tris pH 8, 150 mM NaCl, 20 mM EDTA, 1 mM Na3VO4, 1 mM NaF, a complete protease

inhibitor cocktail (Roche), and anti-phosphatase cocktails 2 and 3 (Sigma-Aldrich). Protein concentrations were quantified with a BCA assay (Bio-Rad). Eighty micrograms of protein were separated by sodium dodecyl sulfate-polyacrylamide and transferred to polyvinylidene difluoride membranes (Millipore). The membranes were blocked with milk or BSA for 1 h before incubation with primary antibodies for 1.5 h. The following mAbs and rabbit polyclonal antibodies were used for immunoblotting: anti-PLC-γ1 (clone D9H10), anti-phosphorylated PLC-γ1 (clone D6M9S), anti-phosphorylated ERK1/2 (clone D13.14.4E), anti-ERK1/2 (clone 137F5), anti-phosphorylated Akt (clone D9E), anti-Akt (clone C67E7), anti-Ku80 (clone C48E7), and anti-phosphorylated tyrosine (clone Tyr-100) purchased from Cell Signaling Technology. The membranes were then washed and incubated with anti-mouse or anti-rabbit horseradish peroxidase-conjugated secondary antibodies from Cell Signaling Technology and GE Healthcare, respectively. Pierce ECL western blotting substrate was used for detection as previously described[64,65]. Quantification was performed using ImageJ software according to the standard gel signal measurement method. Uncropped and unprocessed scans of the blots are available in the Source Data file.

**Cytokine analysis**. Plasma or culture supernatants were collected at the indicated times, and cytokine production (TNF-α, IFN-γ, IL-2, IL-4, IL-6, IL-10, and IL-17A) was determined using the Cytometric Bead Array Th1/Th2/Th17 Kit (BD Biosciences) according to the manufacturer's instructions.

**Liquid chromatography–mass spectrometry metabolite analyses**. At day 16 of culture, $3 \times 1.10^6$ cells of each culture condition were separated from stimulation beads, washed twice in PBS and stored at −80 °C. The extraction solution was composed of 50% methanol, 30% acetonitrile, and 20% water. After the addition of extraction solution, the samples were vortexed for 5 min at 4 °C and then centrifuged at $16,000 \times g$ for 15 min at 4 °C. The supernatants were collected and stored at −80 °C until analyses. Liquid chromatography–mass spectrometry analyses were conducted on a QExactive Plus Orbitrap mass spectrometer equipped with an Ion Max source and a HESI II probe and coupled to a Dionex UltiMate 3000 UPLC system (Thermo Fisher Scientific) as previously described[66]. Data were acquired using Thermo Xcalibur software (Thermo Fisher Scientific). The peak areas of metabolites were determined using Thermo TraceFinder software (Thermo Fisher Scientific), identified by the exact mass of each singly charged ion, and by known retention time on the HPLC column.

**VCN assessment**. Genomic DNA was isolated from transduced cells using a Genomic DNA Purification kit (Qiagen). gDNA was digested with HindIII HF (New England Biolabs, Evry, France) in a total reaction mixture of 6 μL at 37 °C for 30 min. Droplet Digital™ PCR (ddPCR) was performed with TaqMan probes designed to detect a lentiviral sequence (Psi) (Bio-Rad) and a sequence in the human genome (Albumin) (Thermo Fisher Scientific) (sequences shown in Supplementary Table 1). The final reaction mixture contained ddPCR Mastermix (Bio-Rad), forward and reverse primers, probe solutions, and digested gDNA. The sample mixture was transferred to a DG8 cartridge and placed into the QX100 droplet generator (Bio-Rad). Sample droplets were transferred to a 96-well PCR plate, and ddPCR was performed using a SimpliAmp thermal cycler (Thermo Fisher Scientific). The plate was analyzed using a QX200 droplet reader (Bio-Rad). Using the manufacturer's QuantaSoft software (Bio-Rad), the concentration of the target amplicon per unit volume of input for each sample was estimated for both Psi and the Albumin reference gene. The VCN was determined for each sample as follows: VCN = 2 x Psi signal/Albumin signal.

**Quantitative real-time polymerase chain reaction (RT-qPCR)**. CAR transcript expression was assessed by RT-qPCR using primers designed in both scFv and EGFRt regions (sequences shown in Supplementary Table 1). Briefly, mRNA was extracted using the RNeasy Mini Kit® (Qiagen) according to the manufacturer's protocol and subjected to reverse transcription (High Capacity RNA-to-cDNA Master Mix (Applied Biosystems™)). mRNA expression levels were assessed using Fast SYBR™ Green Master Mix (Applied Biosystems™) with a Viia7 thermocycler (Thermo Fisher Scientific). The fold-change for each tested gene was normalized to that of the housekeeping gene HPRT (hypoxanthine guanine phosphoribosyl transferase). The relative expression of each gene was calculated using the $2^{-\Delta\Delta CT}$ method[67].

**RNA sequencing sample preparation**. Fifty-thousand living single-cell expanded Tregs from each condition were directly sorted at Day 16 on a BD FACSAria™ cell sorter, into lysis buffer (TCL Buffer (QIAGEN) supplemented with 1% 2-mercaptoethanol) at a concentration of 1000 cells per 5 μL, and frozen in a DNA Lo-Bind tube (Eppendorf, #022431021) at −80 °C.

Population low-input RNA-seq was then performed from the 5 μL of collected lysis buffer following the standard ImmGen low-input protocol (www.immgen.org). Smart-seq2 libraries were prepared as described previously[68] with slight modifications. Briefly, total RNA was captured and purified on RNAClean XP beads (Beckman Coulter). Polyadenylated mRNA was then selected using an anchored oligo(dT) primer (50 –AAGCAGTGGTATCAACGCAGAGTACT30VN-30) and converted to cDNA

via reverse transcription. First strand cDNA was subjected to limited PCR amplification followed by Tn5 transposon-based fragmentation using the Nextera XT DNA Library Preparation Kit (Illumina). Samples were then PCR amplified for 12 cycles using barcoded primers such that each sample carries a specific combination of eight base Illumina P5 and P7 barcodes for subsequent pooling and sequencing. Paired-end sequencing was performed on an Illumina NextSeq 500 using 2 x 38 bp reads with no further trimming. Reads were aligned to the human genome (GENCODE GRCh38 primary assembly and gene annotations v27) with STAR 2.5.4a (https://github.com/alexdobin/STAR/releases). The ribosomal RNA gene annotations were removed from the GTF file. The gene-level quantification was calculated by featureCounts (http://subread.sourceforge.net/). Raw read counts tables were normalized by median of ratios method with DESeq2 package from Bioconductor (https://bioconductor.org/packages/release/bioc/html/DESeq2.html) and then converted to GCT and CLS formats.

**RNAseq quality control and analysis**

*Quality control*. We screened for contamination by using known cell type-specific transcripts (per ImmGen ULI RNAseq and microarray data). Finally, The RNA integrity for all samples were measured by median Transcript Integrity Number (TIN) across human housekeeping genes with RSeQC software (http://rseqc.sourceforge.net/#tin-py). All samples passed the quality control, including the following criteria: (1) more than 1 million uniquely mapped reads; (2) more than 8,000 genes with over ten reads (3) TIN superior at 45.

*Differential gene expression*. To limit the background noise and avoid unreliable values from low expression or/and high variability intrareplicate, only genes with a minimum average read count of 15 tpm in one of the groups and an inter-replicate coefficient of variation (CV) intrareplicate <0.6 were retained for analysis. We used an uncorrected *t*-test to assess differential gene expression between the different groups from the normalized read counts dataset.

*Signature overlap*. An activation human Treg signature was curated from a published human dataset (GSE76598)[69] downloaded through the Gene Expression Omnnibus (GEO) (https://www.ncbi.nlm.nih.gov/geo/). To reduce noise, genes with an inter-replicate CV intrareplicate <0.7 and an expression level >20 in either comparison groups were selected. Up and downregulated transcripts from the comparison "stimulated versus non stimulated ex vivo native Tregs" were selected at an arbitrary threshold of FC > 1.5 or FC < 0.75, and a *t*-test *p*-value < 0.05. The overlap significance into our dataset was assessed by Chi square test.

**TSDR DNA methylation analysis**. Regulatory T-cell-specific demethylation region (TSDR) DNA methylation analysis was performed as previously described[70] using genomic DNA isolated from freshly sorted or expanded Treg cells using the QIAamp DNA Mini Kit (Qiagen). A minimum of 60 ng bisulfite-treated (EpiTect; Qiagen) genomic DNA was used for RT-qPCR to quantify $\Sigma$ TSDR demethylation. Real-time PCR was performed in a final reaction volume of 20 μL containing 10 μL FastStart universal probe master (Roche Diagnostics), 50 ng/μL lambda DNA (New England Biolabs), 5 pmol/μL methylation or nonmethylation-specific probe, 30 pmol/μL methylation or nonmethylation-specific primers, and 60 ng bisulfite-treated DNA or a corresponding amount of plasmid standard. Samples were analyzed in triplicate on an ABI 7500 cycler (Thermo Fisher Scientific), and the results are reported as % T cells with a demethylated TSDR region.

**XenoGVHD**. All appropriate procedures were performed in the animal facility (registration number A75-15-34) and followed to ensure animal welfare. Eight- to 12-week-old male NSG mice (bred in house or purchased from Charles River) were intraperitoneally injected with 25 mg/kg busulfan (Merck) on days −2 and −1 before injection of $5 \times 10^6$ HLA-A2+ PBMCs into the tail vein with or without $5 \times 10^6$ CAR-Tregs injected retroorbitally via the venous sinus. Saline-injected mice served as controls. CAR-Tregs were generated from three different healthy donors. GVHD was scored based on weight, hunching, fur properties, diarrhea and skin integrity, with 0 to 1 point per category as previously described[71]. On the indicated days and after isoflurane anesthesia supplemented with tetracaine analgesia, peripheral blood from the venous sinus was harvested and centrifuged, and the plasma was collected and frozen before cytokine measurement. Then, the erythrocytes were lysed (RBC lysis buffer, Ozyme), and leukocytes were evaluated by flow-cytometry analysis. When a mouse reached a score of 4, it was sacrificed by cervical dislocation, and the spleen was collected for flow-cytometry analysis, whereas the lungs and liver were harvested for histology. Mouse tissues were fixed in 4% paraformaldehyde and paraffin embedded. Liver and lung sections (4 μm) were stained with hematoxylin and eosin, scanned (Nanozoomer 2.0, Hamamatsu) and blindly assessed by two independent researchers using NDPview software (Hamamatsu).

**Luciferase assay**. To evaluate CAR-Treg homing in vivo, sorted Tregs (CD4+ CD45RO+ CD25hiCD127lo) were activated as described above. Two days later, the cells were transduced with either a CD28 CAR lentivirus or a 4-1BB CAR lentivirus at a MOI of 30 together with luciferase-mCherry-lentivirus at an MOI of 5. The lentiviral plasmid encoding a firefly luciferase protein (pEFS-eFFLY-mCherry) was

kindly provided by Dr. Matthias Titeux. After 7 days of culture, double-transduced mCherry+EGFR+ Tregs expressing the CAR (or MOCK, the negative control) and luciferase were sorted before restimulation as described in Fig. 1c. On day 18 of culture, $5 \times 10^6$ luciferase- Tregs and $5 \times 10^6$ human allogeneic HLA-A2+ PBMCs were injected intravenously into conditioned NSG mice. For bioluminescence imaging, D-luciferin potassium salt (150 mg/kg, Perkin Elmer) was intraperitoneally injected before anesthesia with isoflurane, and images were acquired within 20 min on an IVIS Spectrum CT (Perkin Elmer). Data were analyzed using Living Image software (IVIS Imaging Systems), and the BLI and X-ray superimposition signals were quantified after 3D bioluminescence imaging tomography (DLIT). Using coregistration with the Automatic Mouse Atlas, a region of interest (ROI) corresponding to the spleen was determined, and absolute number of cells was obtained using in vitro calibration with corresponding cells.

**xCELLigence cytotoxicity assay**. To evaluate the cytotoxic potential of CAR-Tregs, the viability of HLA-A2+ and HLA-class I-deficient (through CRISPR-induced deletion) endothelial cell lines (human CiGEnc cell line (11RRID: CVCL_W185), provided by D. Anglicheau) was monitored every 15 min for 10 h by electrical impedance measurement with an xCELLigence RTCA MP instrument (ACEA Biosciences). In each E-plate (ACEA Biosciences) well, $1 \times 10^4$ HLA-A2+ or HLA-class I-deficient endothelial cells were seeded. After 15 h, $2 \times 10^4$ CAR-Tregs or CD8+ CAR-T cells were added to the culture. As a cytotoxicity control, CD8+ T cells were transduced 2 days after activation with the CAR construct at an MOI of 40 and incubated for 18 h for transduction. On day 5 posttransduction, CD8+ EGFRt+ cells were sorted using a FACS Aria II. CD8 CAR-T cells were cultured in complete medium supplemented with IL-2 (100 UI/mL).

The cell indices (CIs) were normalized to the reference value (measured just prior to adding CAR-Tregs to the culture). The normalized cell index in experimental wells was normalized to that of the control wells containing the endothelial cell lines only.

**TRAC deletion using CRISPR/Cas9 genome editing**. gRNA sites in human *TRAC* exonic loci were identified using the online optimized design software at http://crispor.tefor.net/[72]. The highest scoring gRNA, which had no off-target sequences with perfect matches in the human genome, the best predicted efficiency and the nearest coding off-target exonic sites containing at least three mismatched nucleotides was selected and purchased from Thermo Fisher (TrueGuide Synthetic sgRNA, Invitrogen). The *TRAC* CRISPR RNA (crRNA)-targeting sequences included CTCTCAGCTGGTACACGGCA GGG.

**Nucleofection**. On the indicated culture day, cells were resuspended in a Cas9 nuclease/gRNA/nucleofection reagent complex using the P3 Primary Cell 4D-Nucleofector X Kit (Lonza) and underwent nucleofection in a 4D-NucleofectorTM Core Unit + 4D-NucleofectorTM X Unit (Lonza) using the EO 115 program according to the manufacturer's instructions. Cells were then split into prewarmed culture medium at $10^6$ cells/mL with IL-2 (1000 UI/mL) and incubated at 37 °C and 5% $CO_2$ for 24 h before removing stimulation anti-CD3/CD28 beads.

**Statistical analysis**. The results are presented as the mean +/− SEM or median for continuous variables. Frequencies of categorical variables are presented as numbers and percentages. Analyses were performed using GraphPad Prism software (version 8.00; GraphPad Software). For statistical comparisons of in vitro data, we used the nonparametric two-tailed Mann–Whitney test for comparisons of two groups. For survival comparisons, the log-rank test was used. $p$-values < 0.05 were considered significant. Autoscaled metabolic data were normalized by median through Metaboanalyst 5.0 and compared with FDR correction. For radar plots and volcano plots, RStudio (version 3.6.3) and the following R packages were used: dplyr, ggrepel, ggplot2, fmsb and scales packages.

**Ethical approvals**. This study has complied with all relevant ethical regulations. Spleen collection from deceased organ donors was approved by a local Institutional Review Board (Project no.: 2016-12-01; approved on December 5, 2016). Animal procedures were approved by the "Services Vétérinaires de la Préfecture de Police de Paris" and by the "Comité d'Ethique en matière d'Expérimentation Animale Paris Descartes (CEEA 34)" under the number APAFIS#23742-2017091815321774 v9, Université Paris Descartes, Paris, France.

**Reporting summary**. Further information on research design is available in the Nature Research Reporting Summary linked to this article.

## Data availability
The authors declare that all data supporting the findings of this study are available in the article and its Supplementary Information Files, or upon request from the corresponding author. The data underlying all the figures and Supplementary Table 1 and Supplementary Table 2 are provided as a source data file. The transcriptomic data reported in this study have been deposited in the Gene Expression Omnibus (GEO) database under accession no. GSE183598. Metabolomics data have been deposited to the EMBL-EBI MetaboLights database (https://doi.org/10.1093/nar/gkz1019, PMID:31691833) with the identifier MTBLS3121. The complete dataset can be accessed here. Source data are provided with this paper.

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

## Acknowledgements

We thank S. Berissi and colleagues at the Plateforme d'histologie et morphologie du petit animal; O. Pellé and colleagues at the Plateforme de cytométrie; E. Panafieu and colleagues at the Laboratoire d'Expérimentation Animale et Transgénèse of the Structure Fédérative de Recherche Necker, Paris; G. Froment, D. Nègre and C. Costa from the AniRA lenti-vectors production facility at the CELPHEDIA Infrastructure and SFR BioSciences (UMS3444/CNRS/US8/Inserm, ENS Lyon, UCBL); the Etablissement Français du Sang for blood supply from healthy donors; SIRION biotech GmbH for providing LentiBOOST™ and A. De Cian (U1154, CNRS UMR7196, Museum National d'Histoire Naturelle, Paris, France) who kindly provided the Cas9 protein. B.L. was supported by the IMAGINE Institute and the DIM-Thérapie Génique/Région Ile-de-France. A.M. was supported by the Assistance-Publique Hôpitaux de Paris. S.C. and T.B. were supported by the Fondation Emmanuel Boussard. The study was also funded by the Fondation Recherche Medicale (PME20160635834), Fondation Centaure, Fondation Day-Solvay, Sandoz Pharmaceuticals, Agence de la Biomédecine and Agence Nationale de la Recherche (DAISY, ANR-20-CE18-0004). The IVIS spectrum used in this study was purchased with a grant from the DIM-Thérapie Génique/Région Ile-de-France. B.L. created Fig. 6a from Servier Medical Art images, according to the terms of use (smart.servier.com).

## Author contributions

B.L., A.M., S.C., T.B., and J.Z. conceived and designed the experiments and wrote the manuscript. B.L., A.M., S.C., T.B., L.R. performed the experiments and analyzed the data. J.L. performed and analyzed the RNA-seq data. K.V. and B.S. performed the TSDR analysis and analyzed the data. M.T. kindly provided the mCherry-Luciferase plasmid and helped with IVIS Spectrum bioluminescence analysis. T.B., M.D., and H.V. blindly assessed the histological scores. E.S. provided help in the design of the CAR constructs/lentiviral vectors. N.P., D.M., and I.N. provided assistance and insights for immuno-metabolism analysis. E.M. performed immunoblot experiments, and E.M. and S.L. analyzed the data. J.-L.T. performed the Luminex analysis. D.A., C.L., E.M., S.L., M.C., and I.A. contributed to the discussion and manuscript editing.

## Competing interests

The authors declare no competing interests.
