## [Peer Review File · Nature Communications]

Transient mTOR inhibition rescues 4-1BB-CAR-Tregs from tonic signal-induced dysfunctionREVIEWER COMMENTS

Reviewer #1 (Remarks to the Author):

Thank you for the opportunity to review the manuscript "CD28 costimulatory domain protects against tonic signaling-induced functional impairment in CAR-Tregs", I sincerely appreciate the time and effort it took to put this work together. The submitted manuscript reflects a well-designed non-clinical study with carefully chosen and conducted in-vitro and in-vivo experimental assays and models. I have the following suggestions to further strengthen the clarity of the manuscript.

Introduction:

(1) "However, although experimental transplant models indicate that donor-specific Tregs produce greater prevention of graft rejection than polyclonal Tregs⁵, the generation of clinical-grade donor-specific Tregs faces major challenges."

Comment: In addition to the Sagoo paper, I would recommend to also cite/discuss the following study as it demonstrates superior potency compared to polyTregs in vitro and higher functionality with respect to control of allograft injury in a humanized mouse model:

<https://pubmed.ncbi.nlm.nih.gov/24102808/>

(2) "CARs are engineered receptors that consist of an extracellular single-chain variable fragment (scFv), which is derived from..."

Comment: I would suggest to re-write the sentence while be more specific to which ones the authors are referring to. scFv's can be derived from murine or humanized sources, but also synthesized and identified based on phage display library screenings. In addition, natural ligands without any antibody component have been used: for example <https://pubmed.ncbi.nlm.nih.gov/26059190/> And there are more ways, for example computationally designed CARs (<https://www.nature.com/articles/s41587-019-0403-9?proof=true>Here) or multi-antigen targeting CARs etc. etc.

(3) "...derived from the antigen-binding regions of a monoclonal antibody, joined to intracellular T cell signaling domains called cluster of differentiation (CD)3 ζ combined with a costimulatory domain (CSD),..."

Comment: For clarity please also introduce the hinge region/ spacer domain and transmembrane domains as major part of a CAR construct.

(4) "... combined with a costimulatory domain (CSD), most frequently that of either 4-1BB or CD28."

Comment: Keep in mind that this is true for 2nd gen CARs, but already not for 3rd gen CARs as they contain two CSDs or 1st gen CARs who did not have any CSD

(5) "A few pioneering studies have provided proof-of-concept evidence that the Treg response can be redirected toward a donor mismatched antigen."

Comment: From a reader perspective, the transition from the previous paragraph to this one is not really clear. Please try to bridge from previous explanations to this paragraph, as in the end I guess you aim to point towards the CD28 4-1BB CSD topic.

(6) "In the field of oncology, studies have shown that tonic activation of CARs, i.e., signaling irrespective of the presence of the CAR ligand, occurs to varying degrees with most CARs, resulting in baseline activation and ..."

Comment: The authors state that tonic activation occurs to varying degrees. So when does this occur primarily? And at which time point it can be identified? As far as I know, during primary T-cell expansion. Please be more specific and modify the sentence accordingly.

(7) "...resulting in baseline activation and eventually leading to T cell exhaustion with a reduced antitumor effect."

Comment: The authors should also include further effects, such as on differentiation status of T-cells, particularly because they investigated this issue in their study.

(8) "Importantly, the 4-1BB CSD was found to significantly reduce tonic signaling-induced exhaustion and be associated with longer survival than the CD28 CSD."

Comment: Please adjust your statements to clarify the consequence of tonic signaling with respect to CAR technology, i.e. decrease of antigen-specific T-cell response.

(9) "In the present study, we exploited the tonic signaling induced by an anti-HLA-A2 CAR to assess its effect on CAR-Treg phenotype, proliferation, metabolism, signaling and function, according to the

type of CSD.”

Comment: I get to wonder here, do the authors think that effects of tonic signaling might be the same independent of the CAR construct? If it is not the case, how do you think one could assess the effects of tonic signaling, independent from the CAR construct used or define a minimum set of assays to be applied (and which ones)? I think this is worth highlighting in the discussion section.

Results

(1) “A reporter gene encoding truncated epidermal growth factor receptor (EGFRt) was placed behind those a signa virus 2A (T2A).”

Comment: Recommend to add the function of this sequence in the brackets (self-cleaving peptide)

(2) “CD69 (Figure 1D) and Glycoprotein-A Repetitions Predominant (GARP) (not shown) expression was strongly induced in an HLA-A2-specific manner by EGFR-expressing Tregs but not by untransduced EGFR-negative Tregs.”

Comment: Did the authors only check for CD69 to verify activation? Why they did not consider testing for more markers (e.g. CD44, CD25, HLA-DR, CD45RO, CD27)?

(3) “Similarly, normalized CAR expression, as assessed by protein L staining, was not significantly different across the CAR CSDs and T cell subsets (Figure 2D, right panel).”

Comment: Figure 2d, however, shows a trend toward higher expression in 4-1BB. The authors are required to discuss this observation.

(4) “Interestingly, a sizeable population of the 4-1BB CAR-Tregs expressed high levels of FOXP3 and HELIOS along with markers of activation (HLA-DR, 4-1BB, and ICOS)...”

Comment: I don't see such strong difference between 4-1BB and CD28 clusters in Figure 3A, as compared to HLA-DR and 4-1BB. For ICOS I do see blue min expression even stronger in 4-1BB and orange-red max expression in both 4-1BB and CD28. Please comment on this or delete ICOS here.

(5) “Interestingly, a sizeable population of the 4-1BB CAR-Tregs expressed high levels of FOXP3 and HELIOS along with ... and suppressive functions (TIGIT and CTLA-4).”

Comment: Not convincing in my opinion, based on results presented in Figure 3a. The patterns for both show even stronger blue min expression in more areas in 4-1BB than CD28, especially for TIGIT. Also red max expression sizable areas are comparable for TIGIT and CTLA-4 between 4-1BB and CD28 tSNE regions. For TIGIT, based on the total areas presented, I do see even higher expression in CD28 than 4-1BB, because in 4-1BB there is a relatively big area with blue min expression, which is not visible in CD28. The authors need to comment on that issue to explain that discrepancy in data interpretation.

(6) “In contrast to CD28 CAR-Tregs and untransduced Tregs, 4-1BB CAR-Tregs displayed high expression of pS6 five days after TCR stimulation.”

Comment: Reference to Figure 4c is missing here..

(7) “To assess the suppressive capacities of CAR-Tregs in vivo in an antigen-specific manner, we used a xenogeneic GVHD model based on the transfer of peripheral blood mononuclear cells (PBMCs) into busulfan-conditioned NSG mice.”

Comment: Abbreviation introduced here for the first time. Please explain, use either “NOD scid gamma (NSG) mice” or “NOD.Cg-Prkdc^{scid} Il2rg^{tm1Wjl} / SzJ (NSG) mice”.

(8) Different doses of hPBMCs were tested (5, 10 or 20 x 10⁶ hPBMCs), demonstrating a dose-dependent effect on both GVHD score and survival (Figure supp 2A-C).”

Comment: Not only GvHD score and survival but also engraftment since you are citing supplementary Figure 2 A-C, where 2c provided data in hCD45 for engraftment analyses.

(9) This precaution stemmed from a previous finding showing that mixing HLA-A2+ PBMCs and HLA-A2-targeted CAR-Tregs before infusion impedes CAR-Treg homing (not shown).

Comment: This is an important finding. Can you provide data in the supplementary if possible to support this?

(10) “We concluded that both 4-1BB CAR-Tregs and CD28 CAR-Tregs were efficient at preventing xenogeneic acute GVHD through early inhibition of hPBMC expansion.”

Comment: It should be pointed out here that 4-1BB CAR-Tregs were additionally stabilized by rapamycin and vitamin C.

(11) “Although Treg exhaustion is still ill-defined, we hypothesized that...”

Comment: What do the authors mean by “ill-defined”? Please specify. Do you mean because several of the canonical exhaustion makers are also markers of Tregs, or simply of activated T cells? Or

because Tregs do not produce inflammatory cytokines, loss of cytokine production cannot be monitored as a surrogate marker?

(12) "Taken together, these findings suggest that 4-1BB CAR tonic signaling promotes accelerated dysfunction in Tregs, consistent with the lack of ability to be restimulated through antigen receptors."

Comment: "Since "tonic signaling-related functional impairment" is the main issue under investigation, the authors could have performed additional analyses such RNA seq and in-depth transcriptome analysis.

Discussion

Please add a paragraph discussing the limitations of the study.

Please also discuss:

1. The potential effects of your findings with respect to other CAR designs or other indications.
 2. Other components of CAR design (hinge/spacer, transmembrane, ecto domain) and their potential contribution to tonic signaling/ chronic T-cell activation effects.
 3. As it was shown that the order of co-stimulatory domains also influence effector functions and degree of functionality/efficacy (section 5.2), the authors should comment on this observation and potential impacts on their research question.
 4. In the presented study, the authors tested 2nd gen CARs with one CSD, either CD28 or 4-1BB. Please comment on potential effects of 3rd gen CARs (two CSDs) or other CAR designs in this context.
 5. In the results section, the authors state that they "measure CAR expression, independent of the ability of each CAR to bind HLA-A2, as this could be a confounding variable across the different CARs" Could you further comment on and discuss other potential confounding variables.
 6. The authors are encouraged to discuss future implications of their findings on the CAR-T field
- Methods

(1) "The MFIs of the tagged scFvs were measured by FACS analysis."

Comment: Which device was used? Please add.

To avoid confusion, I advise the authors to change FACS to "flow cytometric analysis" throughout the method section, because FACS is more commonly used when referring to cell sorting.

(2) "...as assessed by anti-HLA-A2/A28 antibody (OneLambda) staining evaluated by FACS analysis."

Comment: Which device was used? Please add.

(3) "Naïve regulatory T cells (nTregs), which were defined as CD4+ CD25++ CD45RA+ CD127low, were sorted using a FACSAria II (BD Biosciences)."

Comment: In the main text, the authors provided the following information for nTregs: "CD4+ CD25Hi CD127- CD45RA+ CD45RO-". Please be consistent throughout the entire manuscript and do not use different styles of writing or order of markers.

(4) "Intracellular staining was performed for FOXP3, HELIOS (Thermo Fisher Scientific), Granzyme B and Ki67 (Thermo Fisher Scientific)."

Comment: no need to repeat the supplier here

Several abbreviations such as SDS-PAGE, PVDF, HRP are mentioned for the first time and not introduced before. Please correct accordingly.

Reviewer #2 (Remarks to the Author):

This manuscript compares CD28 and 4-1BB containing second generation CARs when expressed in Tregs. While interesting and technically generally well executed, I have substantial reservations that temper my enthusiasm.

Major issues

1. My major concern is the somewhat incremental nature of the work, given previous comparisons of second generation CARs in Tregs in which different co-stimulatory domains were used. See Dawson et al Sci Transl Med in particular (ref 11 in the manuscript). This study systematically compared a panel of 10 CARs that employ different co-stimulatory units and showed - superior persistence and efficacy of the CD28 containing variant

- proliferation and Treg lineage instability linked to the 4-1BB containing CAR variant
These findings compromise novelty of the present study.

2. It might be argued that novelty of the study relates to the identification of tonic signaling in a CAR Treg context. However, I also consider this incremental given the fact that tonic signaling in a well recognised phenomenon in Tconv T-cells. Moreover, tonic signaling mediated by 4-1BB-containing second generation CARs has been described in Tconv cells previously by Milone et al (ref 44 in the manuscript) and also by Gomes-Silva et al (ref 26 in the manuscript). Outcomes were markedly different in these two examples such that activated CAR T-cell proliferated constitutively in the former example, while activation induced cell death occurred in the latter case. Here, we see a broadly similar form of tonic signaling to that described by Milone.

3. Specifically, Milone et al observed CAR ligand-independent proliferation of CD19-specific CAR T-cells (presumed Tconv) of both the CD4+ and CD8+ T cell lineage. This was "associated with a prolonged blast phase after the initial stimulation and transduction". Milone concluded that "the enhanced growth effects of the α CD19-BB- ζ receptor are consistent with the antigen-independent growth effects that are observed in T cells stimulated through the natural CD137 receptor by agonist monoclonal antibody." These results are broadly similar to those reported here in Tregs. It would be useful to clarify if this effect is dependent on recent activation through the TCR? (as occurs on d11 according to the scheme shown in Fig. 1c).

4. The introductory section over-simplifies what is known about tonic signaling. Multiple forms of tonic signaling have been described in Tconv T-cells engineered to express either CD28 or 4-1BB containing CARs. However, the authors preferentially link tonic signaling to CD28-containing CARs with statements in the Introduction (first two) and Discussion (third statement) such as
- "Tonic signaling may cause CAR-T cell dysfunction, especially when the CAR structure incorporates the CD28 costimulatory domain (CSD) rather than the 4-1BB CSD."
- "This study demonstrates that CD28-CAR best preserves Treg function and survival in the context of tonic signaling, in contrast with previous findings for Tconvs."
- "Strikingly, although CD28 was found to worsen the process of tonic signaling-induced exhaustion in Tconvs, our data showed the opposite in Tregs."

The undue emphasis on CD28-associated tonic signaling seems to imply that the demonstration here of 4-1BB tonic signaling in a Treg is a highly novel finding rather than something that might be expected, given the full breadth of the prior literature.

5. Tonic signaling was not characterised by unregulated expression of CD69 or constitutive cytokine production. Why is only HLA-DR upregulated? Did authors test for tonic signaling using other assays (eg constitutive phosphorylation of CD3z; Jurkat Nur77-RFP reporter cells (Smith et al Sci Transl Med 2019: Vol. 11, Issue 485, eaau7746)).

6. Authors comment on the affinity of scFvs but no formal affinity data are presented (e.g. Kd, on rates, off rates etc).

7. Authors demonstrate higher expression of 4-1BB CAR than CD28 CAR, indicated by higher pentamer and protein L binding. Is this due to enhanced transcription, given the link between 4-1BB tonic signaling and promoter selection (Gomes-Silva reference)?

8. Co-expression of CAR and EGFRt reporter should theoretically be stoichiometric given the use of a 2A peptide. In keeping with this, authors state that "the mean fluorescence intensities (MFIs) of HLA-A2 pentamer and EGFR staining were not different across the CAR CSDs and cell lineages (Figure 2D)." To this reviewer however, there appears to be a trend towards higher expression of the 4-1BB CAR (indicated by higher pentamer and protein L binding), accompanied by a trend towards reduced EGFRt expression by 4-1BB CAR Tregs (all shown in Fig. 2d). Further replicate experiments and expression of the data as a ratio between CAR expression to EGFRt expression might unmask a significant difference in this respect. This underpins the need for transcriptional data.

9. Authors infer that 4-1BB tonic signaling was primarily seen in Tregs. How was tonic signaling excluded in Tconv that expressed these CARs.
10. What is the evidence that MAPK activation drives 4-1BB mediated proliferation.
11. I may have missed this but I could not identify the supplementary Tables.
12. While the Dawson citation compromises novelty, a number of differences emerge between the two studies. For example Dawson reported preservation of high levels of central memory Treg by 4-1BB CAR Tregs whereas effector differentiation was more evident here. Please discuss differences in findings regarding 4-1BB CAR Tregs between the two studies.
13. The analysis of immunometabolism is cursory, incorporating only a limited number of genes.
14. What were the kinetics of kinetics of s6 phosphorylation in 4-1BB CAR Tregs after activation on d11?
15. It is stated that "In line with these findings, the expansion rate of 4-1BB CAR-Tregs decreased to a level comparable to those of CD28 CAR-Tregs and untransduced Tregs (Figure 5D)." However, I do not see the CD28 and untransduced Treg controls in Figure 5D.
16. Could in vivo persistence of Tregs in the present study be linked to the use of a different conditioning regimen to that used in other studies?
17. Why is pulmonary transit of Treg not evident in figure 7?
18. Authors comment that their findings conflict with two recent studies reporting the lack of in vivo suppressive function of 4-1BB CAR-Tregs. Could this be due to the fact that the Treg-supporting cocktail was added to the culture medium from days 10 to 18 for 4-1BB CAR-Tregs prior to in vivo testing?
19. Authors comment that "4-1BB CAR-Tregs potently suppressed xenoGVHD despite a reduced ability to be stimulated through the CAR. This finding is reminiscent of a previous report showing in vivo antigen-specific suppression mediated by HLA-A2-targeted CAR-Tregs, even in the absence of signaling domains in the CAR structure." Is this not likely to be due to the ability of signaling defect CARs to facilitate retention of these Tregs at the site of the target antigen (HLA-A2).

Minor points

1. It is incorrect to state that CARs are engineered receptors that consist of an extracellular single-chain variable fragment (scFv)... Many CARs employ alternative ligand binding moieties to scFvs.
2. In the cartoons shown in Fig 1a, it would be helpful to include additional structural details of CARs eg hinge/ spacer and transmembrane domain.
3. A reference citation should be provided for EGFRt (Jensen et al).
4. Please specify the promoter used in the LV vector, given the known importance of promoter choice in 4-1BB tonic signaling (Gomes-Silva reference).
5. A dotted line symbol should be included in the legend for Fig 1e (NT cells).
6. It is preferable not to use red and green colors in figures, in consideration of color blind individuals.

Reviewer #3 (Remarks to the Author):

The authors of this manuscript aimed to study the tonic signalling in Tregs transduced with CAR constructs. They compared Tregs transduced with two CAR constructs both specific for the same antigen (HLA-A2) but with two different costimulatory endodomains. They demonstrated that the Tregs expressing the CAR with the 4-1BB, compared to CD28, proliferated more, had greater

activation of MAP kinase and mTOR pathways, but decreased stability and capacity to be restimulated via the CAR.

This paper addresses a novel question, namely the effect of the costimulatory endodomanins in the tonic signal in Tregs. This question has been addressed before with conventional T cells. Altogether this is an interesting and well performed study showing very clearly that for Tregs CD28 reduced tonic signalling compared to 4-1BB, opposite of what was seen for conventional T cells. However, I have a series of comments that are listed below.

1. One reservation that I have is that the authors did not compare their results with what was published by Boroughs et al in 2019 in JCI Insight. Although the aims of the two studies were different some analysis of the Tregs are in common but the results are different. The authors need to add this reference and compare their data with the results from Boroughs study.
2. Linked to the previous point in the Introduction (line 107) the authors need to cite other publications where the negative effect of 41BB was explored for Tregs.
3. The authors need to clarify in the text the way that the Tregs were stimulated for each result reported. For example, in Figure 1e the authors wrote in the Figure legend that the Tregs were 'TCR restimulated'. Is this an allogeneic stimulation of splenocytes expressing HLA-A2? If this is the case both TCR and CAR are engaged, is this correct? The authors need to clarify this point and where possible separate the TCR stimulation from the CAR. The authors should at least comment on this point.
4. The authors speculate on lines 169-171 that 'ligand-independent CAR tonic signalling seems to primarily dependent on the high cell-surface density of the CAR', which altogether entail CAR clustering'. The authors speculate that the high expression density of 41BB seen with pentamers correspond to a cluster of CAR molecules. However, this should be demonstrated directly using other approaches such as confocal microscopy. In addition, the authors say that proliferation is induced by endogenous TCR, and linked to the previous point does this mean that they used splenocytes HLA-A2 negative?
5. The author report on lines 172-175 that 'the vector copy number (VCN) in transduced Tregs was similar between CD28 CAR and 4-1BB CAR'. However, the VCN is not enough to demonstrate that the number of CD28 and 41BB molecules are the same. For instance, the mRNA of 41BB might be more stable. The level of total 41BB or CD28 protein should be evaluated by western blot or other approaches to quantify the amount of target protein normalised to the total cell proteins.
6. The authors need to modify their statement in line 176 as it is not fully correct. Data only show a higher expression of 41BB molecule on the cell surface and not necessarily a greater vector expression (VCN similar) and higher protein production (see comment above).
7. On line 188-191, the authors contradict themselves. On line 169 they suggest that the tonic signal of 41BB is due to the higher expression of CAR molecule. However, using protein L they show CAR expression is similar in both in CD28 and 41BB CAR cells. This point needs to be clarify. The statement "these results suggest that constitutive 41BB stimulation itself interferes with human Treg biology" is not clear and should be explained more in detail.
8. The phosphorylation of the MAP kinase and AKT pathways has been observed after 5 days of stimulation with beads from what is shown in Figure 1c (is this correct)? What happen after few more days? Is the phosphorylation maintained at the same level or decreases?
9. In the p-tyrosine blot (Fig 2 e), did the authors investigate the differential expression of tyrosine phosphorylation in 41BB and CD28 CAR Treg cells at just below 125 (41BB) or above 50 (CD28) kDa? That seems to be particularly relevant to understand the different signals induced by the two CSDs.
10. The CD28 and 41BB CAR Treg cells in the tSNE analysis do not seems highly clustered as stated by the authors (line 225-228). Although there is a clearly different clusterisations, all the markers shown in Fig. 3a, with the exception of HLA-DR and CD15s, seem to form clusters in both populations. The authors should also comment about the over-expression of 41BB in 41BB CAR Treg cells. Can the CAR signal self-support its own expression? The authors should also stress that the tSNE analysis shows difference in the expression and not necessarily identify positive and negative cell populations.
11. The production of cytokines in Figure 3b shows a difference with the study from Boroughs for

TNF- α , the authors need to comment on this.

12. In Figure 5, the authors report the culturing the 41BB Tregs with Rapa/ VitC on different parameters. However the authors should present additional data presented in Figure 1-4 before using the Rapa/VitC in vivo.

13. For the experiment in vivo, I do not understand why the authors have tested only the 41BB Tregs treated with Rapa/VitC as the entire aim of the study was to compare the two co-stimulatory endo-domains in the absence of any additional factors that can modify the behaviour of the Tregs. In my view (although I realise it is a lot of work) these two Treg lines (in the absence of Rapa/VitC) should be compared.

14. It is not clear to me but I imagine that the 41BB Tregs in Figure 8 were not treated with Rapa/VitC. If this is correct more reason for the 41BB Tregs in the in vivo experiment to be untreated. At least the author should compare in the experiment described in Figure 8 the 41BB Tregs treated and not with Rapa/VitC.

Minor

1. Can the authors describe in the MM what the HLA-A2/A28-negative cytopheresis kits is?
2. There is a mistake on line 143 the authors wrote HLA-A28 negative, should be HLA-A2 negative.

REVIEWER COMMENTS

Reviewer #1 (Remarks to the Author):

Thank you for the opportunity to review the manuscript “CD28 costimulatory domain protects against tonic signaling-induced functional impairment in CAR-Tregs”, I sincerely appreciate the time and effort it took to put this work together. The submitted manuscript reflects a well-designed non-clinical study with carefully chosen and conducted in-vitro and in-vivo experimental assays and models. I have the following suggestions to further strengthen the clarity of the manuscript.

We are thankful for this positive feedback and insightful guidance. The revised manuscript has been extensively updated according to these suggestions.

Introduction:

(1) “However, although experimental transplant models indicate that donor-specific Tregs produce greater prevention of graft rejection than polyclonal Tregs⁵, the generation of clinical-grade donor-specific Tregs faces major challenges.”

Comment: In addition to the Sagoo paper, I would recommend to also cite/discuss the following study as it demonstrates superior potency compared to polyTregs in vitro and higher functionality with respect to control of allograft injury in a humanized mouse model:

<https://pubmed.ncbi.nlm.nih.gov/24102808/>

We changed the sentence and incorporated the citation as follows:

Page 4: “However, although experimental transplant models including in highly relevant humanized mouse models^{4,5}, large-scale ... challenges”.

(2) “CARs are engineered receptors that consist of an extracellular single-chain variable fragment (scFv), which is derived from...”

Comment: I would suggest to re-write the sentence while be more specific to which ones the authors are referring to. scFv’s can be derived from murine or humanized sources, but also synthesized and identified based on phage display library screenings. In addition, natural ligands without any antibody component have been used: for example <https://pubmed.ncbi.nlm.nih.gov/26059190/> And there are more ways, for example computationally designed CARs (<https://www.nature.com/articles/s41587-019-0403-9?proof=trueHere>) or multi-antigen targeting CARs etc. etc.

(3) “...derived from the antigen-binding regions of a monoclonal antibody, joined to intracellular T cell signaling domains called cluster of differentiation (CD)3ζ combined with a costimulatory domain (CSD),...”

Comment: For clarity please also introduce the hinge region/ spacer domain and transmembrane domains as major part of a CAR construct.

(4) “... combined with a costimulatory domain (CSD), most frequently that of either 4-1BB or CD28.”

Comment: Keep in mind that this is true for 2nd gen CARs, but already not for 3rd gen CARs as they contain two CSDs or 1st gen CARs who did not have any CSD.

A few sentences in the introduction have been rewritten according to the Reviewer’s suggestions to provide a more comprehensive and extensive description of CAR architecture:

Page 4: “CARs are engineered ... in the clinics.”

(5) “A few pioneering studies have provided proof-of-concept evidence that the Treg response can be redirected toward a donor mismatched antigen.”

Comment: From a reader perspective, the transition from the previous paragraph to this one is not really clear. Please try to bridge from previous explanations to this paragraph, as in the end I guess you aim to point towards the CD28 4-1BB CSD topic.

A few sentences in the introduction have been rephrased to address the Reviewer’s comments:

Pages 4-5: “Although CAR technology ... suppressive function^{7,11,19”}

(6) “In the field of oncology, studies have shown that tonic activation of CARs, i.e., signaling irrespective of the

presence of the CAR ligand, occurs to varying degrees with most CARs, resulting in baseline activation and ...”
Comment: The authors state that tonic activation occurs to varying degrees. So when does this occur primarily? And at which time point it can be identified? As far as I know, during primary T-cell expansion. Please be more specific and modify the sentence accordingly.

The following sentence in the Discussion section provides hints and suggestions to identify tonic signaling in CAR-Tconvs and CAR-Tregs:

Page 24: “Although, there are currently ... of 4-1BB signaling.”

(7) “...resulting in baseline activation and eventually leading to T cell exhaustion with a reduced antitumor effect.”
Comment: The authors should also include further effects, such as on differentiation status of T-cells, particularly because they investigated this issue in their study.

(8) “Importantly, the 4-1BB CSD was found to significantly reduce tonic signaling-induced exhaustion and be associated with longer survival than the CD28 CSD.”

Comment: Please adjust your statements to clarify the consequence of tonic signaling with respect to CAR technology, i.e. decrease of antigen-specific T-cell response.

A more general picture of the complexity of CAR tonic signaling has been included in the revised manuscript. We also better emphasize the issues related to tonic signaling specifically in Tregs. The respective role of CD28 and 4-1BB CSD in Tconvs and Tregs has been moved to the discussion section to avoid an overly long introduction.

Pages 5-6 (introduction): “CAR tonic signaling ... type of CSD”.

Page 21 (discussion) “In regard to tonic signaling ... CSD tonic signal in Tconv.”

(9) “In the present study, we exploited the tonic signaling induced by an anti-HLA-A2 CAR to assess its effect on CAR-Treg phenotype, proliferation, metabolism, signaling and function, according to the type of CSD.”

Comment: I get to wonder here, do the authors think that effects of tonic signaling might be the same independent of the CAR construct? If it is not the case, how do you think one could assess the effects of tonic signaling, independent from the CAR construct used or define a minimum set of assays to be applied (and which ones)? I think this is worth highlighting in the discussion section.

We are grateful for these suggestions. In the revised introduction (see above), we explain that CAR tonic signaling involves a wide range of mechanisms, with varying ensuing effects. Unfortunately, there are no simple or universal rules for predicting the impact of CAR tonic signaling on T cell function. Therefore, each CAR construct has to be tested empirically. The revised manuscript now issues some recommendations in the discussion section with regard to the importance of tonic signaling assessment in CAR Tregs.

Page 24 “These data highlight ... of 4-1BB signaling.”

Results

(1) “A reporter gene encoding truncated epidermal growth factor receptor (EGFRt) was placed behind thosea asigna virus 2A (T2A).”

Comment: Recommend to add the function of this sequence in the brackets (self-cleaving peptide)

This change was made accordingly (Page 7).

(2) “CD69 (Figure 1D) and Glycoprotein-A Repetitions Predominant (GARP) (not shown) expression was strongly induced in an HLA-A2-specific manner by EGFR-expressing Tregs but not by untransduced EGFR-negative Tregs.” Comment: Did the authors only check for CD69 to verify activation? Why they did not consider testing for more markers (e.g. CD44, CD25, HLA-DR, CD45RO, CD27)?

CD69 (Figure 1) and GARP (Figure S2) were assessed as early activation markers following cognate antigen (HLA A2) stimulation. These two markers have been broadly used to assess transient activation of human Tregs *in vitro*. Furthermore, CD69 was demonstrated to be upregulated upon activation in both *ex vivo* and cultured Tregs (Issa F et al. Front Immunol 2019). In contrast, high levels of CTLA-4, TIGIT, CD25, and GITR expression are maintained in cultured Tregs (Issa F et al. Front Immunol 2019) and would be of less interest for addressing the question of antigen-specific activation.

However, later in culture, on day 16, 4-1BB-CAR-Tregs failed to express CD69 in response to CAR stimulation (Figure 8). Hence, we tested for a broader panel of activation markers. HLA-

DR was strongly and constitutively induced by the 4-1BB tonic signal and was not useful for testing the response to CAR stimulation (Figure 8). In contrast, CAR-Tregs demonstrated the ability to enhance CD25 and 4-1BB expression upon stimulation with cognate antigen.

(3) “Similarly, normalized CAR expression, as assessed by protein L staining, was not significantly different across the CAR CSDs and T cell subsets (Figure 2D, right panel).”

Comment: Figure 2d, however, shows a trend toward higher expression in 4-1BB. The authors are required to discuss this observation.

This is a well-taken point. This set of data has been moved to Figure S4 in the revised manuscript, and the trend toward greater CAR expression is now acknowledged in the Results section (p. 11). However, we do not believe that the strong 4-1BB tonic signaling in Tregs results merely from a greater expression of the 4-1BB CAR transgene than of the CD28 CAR transgene for the following reasons:

1- The vector copy numbers were roughly similar across CAR constructs and CD4 cell lineages (Fig. 2h).

2- A bicistronic lentiviral vector based on T2A cleaving peptide expresses the two proteins (CAR and EGFRt) in a stoichiometric manner. In this respect, comparison of cell size-normalized EGFRt expression between 4-1BB and CD28 CARs revealed opposite trends (Figure S4a).

3- We studied CAR expression through RT-PCR using two different sets of primers that amplified ScFv and EGFR in four independent CAR-Treg cultures. These experiments showed that CAR transcript expression was highly similar between the two constructs (Fig. 2i).

However, we were unable to exclude that the tonic signal itself and ensuing mTORC1 activation promote lysosomal dysfunction and CAR accumulation at the cell membrane (Jin J et al. Sci Immunol 2021, Li W et al. Immunity 2020).

Therefore, these findings were discussed as follows:

Page 11: “From these observations, we inferred that 4-1BB ... at the cell membrane.”

(4) “Interestingly, a sizeable population of the 4-1BB CAR-Tregs expressed high levels of FOXP3 and HELIOS along with markers of activation (HLA-DR, 4-1BB, and ICOS)...”

Comment: I don't see such strong difference between 4-1BB and CD28 clusters in Figure 3A, as compared to HLA-DR and 4-1BB. For ICOS I do see blue min expression even stronger in 4-1BB and orange-red max expression in both 4-1BB and CD28. Please comment on this or delete ICOS here.

(5) “Interestingly, a sizeable population of the 4-1BB CAR-Tregs expressed high levels of FOXP3 and HELIOS along with ... and suppressive functions (TIGIT and CTLA-4).”

Comment: Not convincing in my opinion, based on results presented in Figure 3a. The patterns for both show even stronger blue min expression in more areas in 4-1BB than CD28, especially for TIGIT. Also red max expression sizable areas are comparable for TIGIT and CTLA-4 between 4-1BB and CD28 tSNE regions. For TIGIT, based on the total areas presented, I do see even higher expression in CD28 than 4-1BB, because in 4-1BB there is a relatively big area with blue min expression, which is not visible in CD28. The authors need to comment on that issue to explain that discrepancy in data interpretation.

We apologize if the above-cited sentence was misleading. We did not intend to claim that the entire 4-1BB CAR-Treg population displayed greater expression of ICOS or TIGIT than its CD28 counterpart. We pointed out a small subset among the 4-1BB population, now delineated with a dashed oval, that fulfilled the phenotypic criteria of effector Tregs (high expression of ICOS, TIGIT, CTLA-4, HLA DR).

These sentences have been rephrased in the Results and Discussion sections:

Page 12 (results section): “The CD28 CAR-Tregs ... met the phenotypic criteria of effector Tregs⁴⁰”

Page 22 (discussion section): “The transcription factor MYB ... Tregs at day 16 of culture⁴⁰”

(6) “In contrast to CD28 CAR-Tregs and untransduced Tregs, 4-1BB CAR-Tregs displayed high expression of

pS6 five days after TCR stimulation.”
Comment: Reference to Figure 4c is missing here.

Page 9: A reference to Figure 4c (Revised Figure 2b) was added to the corresponding results section.

(7) “To assess the suppressive capacities of CAR-Tregs *in vivo* in an antigen-specific manner, we used a xenogeneic GVHD model based on the transfer of peripheral blood mononuclear cells (PBMCs) into busulfan-conditioned NSG mice.”

Comment: Abbreviation introduced here for the first time. Please explain, use either “NOD scid gamma (NSG) mice” or “NOD.Cg-Prkdc^{scid} Il2rg^{tm1Wjl} / SzJ (NSG) mice”.

Page 15: The mouse strain is now properly spelled when first introduced.

(8) Different doses of hPBMCs were tested (5, 10 or 20 x 10⁶ hPBMCs), demonstrating a dose-dependent effect on both GVHD score and survival (Figure supp 2A-C).”

Comment: Not only GvHD score and survival but also engraftment since you are citing supplementary Figure 2 A-C, where 2c provided data in hCD45 for engraftment analyses.

Page 15: This sentence has been changed accordingly “Different doses ... and survival (Figure Supp 6a-c).”

(9) This precaution stemmed from a previous finding showing that mixing HLA-A2+ PBMCs and HLA-A2-targeted CAR-Tregs before infusion impedes CAR-Treg homing (not shown).

Comment: This is an important finding. Can you provide data in the supplementary if possible to support this?

Cazaux M et al. previously showed that the binding of CD19-specific CAR-T cells to their CD19+ target cells immediately after their administration strongly limits their ability to recirculate through lymphoid organs. This finding raised the concern that mixing HLA A2-targeted CAR-Tregs and HLA A2+ PBMCs before infusion would significantly impede their circulation. Hence, we performed two preliminary experiments (a total of 5 mice in each group), where CAR-Tregs and PBMCs were administered either separately or together. Based on the results depicted in the figure below, we opted for separate administration, even though the difference fell short of statistical significance, likely due to low sampling size.

(10) “We concluded that both 4-1BB CAR-Tregs and CD28 CAR-Tregs were efficient at preventing xenogeneic acute GVHD through early inhibition of hPBMC expansion.”

Comment: It should be pointed out here that 4-1BB CAR-Tregs were additionally stabilized by rapamycin and vitamin C.

This is a very well taken point. Additional experiments were performed to add a group of untreated 4-1BB CAR-Tregs. We now demonstrate that unduly activated CAR-Tregs, due to 4-1BB tonic signaling, exhibited reduced *in vivo* suppressive capacities and survival compared to CD28 CAR-Tregs. Notably, the Treg-friendly cocktail, which enforced metabolic rest, improved early expansion, suppressive function and *in vivo* persistence. This is now clearly indicated and properly discussed in the revised manuscript:

Page 16: “Regarding survival, all the mice ... and related death (Figure 6b).”

Page 23: “More importantly, the addition ... multikinase inhibition⁵⁸.”

(11) “Although Treg exhaustion is still ill-defined, we hypothesized that...”

Comment: What do the authors mean by “ill-defined”? Please specify. Do you mean because several of the canonical exhaustion makers are also markers of Tregs, or simply of activated T cells? Or because Tregs do not produce inflammatory cytokines, loss of cytokine production cannot be monitored as a surrogate marker?

The following sentence has been added to the discussion to clarify this important point:

Page 24: “Notably, increased tonic signal ... 4-1BB signaling.”

(12) “Taken together, these findings suggest that 4-1BB CAR tonic signaling promotes accelerated dysfunction in Tregs, consistent with the lack of ability to be restimulated through antigen receptors.”
Comment: “Since “tonic signaling-related functional impairment” is the main issue under investigation, the authors could have performed additional analyses such RNA seq and in-depth transcriptome analysis.

We are grateful for this suggestion. We performed RNAseq analysis in two independent CAR-Treg lines at day 16 of culture: untransduced, CD28-CAR-Tregs, and 41BB-CAR-Tregs.

These results are depicted in revised Figure 3b-e

Discussion

(1) Please add a paragraph discussing the limitations of the study.

The following paragraph has been added to the discussion:

Pages 24-25: “This study has one primary limitation. ... for future investigations.”

Please also discuss:

- (1). The potential effects of your findings with respect to other CAR designs or other indications.
- (6). The authors are encouraged to discuss future implications of their findings on the CAR-T field

The discussion now extensively covers the implications of our findings, especially the following:

- the importance of assessing the effect of potential CAR tonic signal for each CAR construct in every lymphoid lineage. Conclusions previously drawn in CAR-Tconv should not be taken for granted in CAR-Tregs.

- the strategies to cope with CAR tonic signaling depends on the identified mechanism. We showed that mTOR inhibitor-induced metabolic rest was efficient at decreasing the negative impact of 4-1BB tonic signaling. We also discuss the possibility of lengthening the distance between 4-1BB CSD and the cell membrane.

Pages 23-24: “We propose that transient mTOR inhibition ... a 3rd generation CAR²¹.”

(2). Other components of CAR design (hinge/spacer, transmembrane, ecto domain) and their potential contribution to tonic signaling/ chronic T-cell activation effects.

This is further discussed in the revised introduction and discussion:

Page 5: “CAR tonic signaling may be ... to self-aggregate²⁵.”

Page 20: “In regard to tonic ... have been described.”

(3). As it was shown that the order of co-stimulatory domains also influences effector functions and degree of functionality/efficacy (section 5.2), the authors should comment on this observation and potential impacts on their research question.

(4). In the presented study, the authors tested 2nd gen CARs with one CSD, either CD28 or 4-1BB. Please comment on potential effects of 3rd gen CARs (two CSDs) or other CAR designs in this context.

The importance of the relative position of the 4-1BB CSD in the 4-1BB tonic signal is now better discussed.

Pages 23-24: “It was also proposed ... in a 3rd generation CAR²¹.”

(5). In the results section, the authors state that they “measure CAR expression, independent of the ability of each CAR to bind HLA-A2, as this could be a confounding variable across the different CARs” Could you further comment on and discuss other potential confounding variables.

We are not certain that we understand this point.

Methods

(1) “The MFIs of the tagged scFvs were measured by FACS analysis.”

Comment: Which device was used? Please add.

To avoid confusion, I advise the authors to change FACS to “flow cytometric analysis” throughout the method section, because FACS is more commonly used when referring to cell sorting.

This point has been clarified, and FACS analysis was replaced with flow cytometry analysis throughout the manuscript.

Page 27: “The MFIs of the tagged scFvs bound ... instrument (Luminex™).”

(2) “...as assessed by anti-HLA-A2/A28 antibody (OneLambda) staining evaluated by FACS analysis.”

Comment: Which device was used? Please add.

This sentence has been changed as follows:

Page 27: “as assessed by anti-HLA-A2/A28 antibody ... using the BD LSRFortessa™ X-20 analyzer”

(3) “Naïve regulatory T cells (nTregs), which were defined as CD4+ CD25++ CD45RA+ CD127low, were sorted using a FACSAria II (BD Biosciences).”

Comment: In the main text, the authors provided the following information for nTregs: “CD4+ CD25Hi CD127- CD45RA+ CD45RO-“. Please be consistent throughout the entire manuscript and do not use different styles of writing or order of markers.

The same phenotypic description of naïve Tregs is now used throughout the manuscript.

(4) “Intracellular staining was performed for FOXP3, HELIOS (Thermo Fisher Scientific), Granzyme B and Ki67 (Thermo Fisher Scientific).”

Comment: no need to repeat the supplier here

This repetition has been removed accordingly.

(5) Several abbreviations such as SDS-PAGE, PVDF, HRP are mentioned for the first time and not introduced before. Please correct accordingly.

The above-cited abbreviations are now properly introduced when they first appear in the manuscript.

Reviewer #2 (Remarks to the Author):

This manuscript compares CD28 and 4-1BB containing second generation CARs when expressed in Tregs. While interesting and technically generally well executed, I have substantial reservations that temper my enthusiasm.

Major issues

(1). My major concern is the somewhat incremental nature of the work, given previous comparisons of second-generation CARs in Tregs in which different co-stimulatory domains were used. See Dawson et al Sci Transl Med in particular (ref 11 in the manuscript). This study systematically compared a panel of 10 CARs that employ different co-stimulatory units and showed

- superior persistence and efficacy of the CD28 containing variant
- proliferation and Treg lineage instability linked to the 4-1BB containing CAR variant

These findings compromise novelty of the present study.

We more than agree with Reviewer #2. Dawson et al. Sci. Transl. Med. 2020 publication is a landmark paper in the field of CAR-Tregs, whose results are abundantly discussed in our manuscript. Indeed, our work confirms and extends some of the conclusions drawn in their study, including the following:

- superior persistence and efficacy of the CD28-containing variant
- greater proliferation and Treg instability related to the 4-1BB-containing CAR variant.

However, our study stresses the deleterious effect of tonic signaling in CAR-Tregs and provides new insights into CAR-Treg biology. Our work addresses for the first time the impact of CAR tonic signaling in Tregs according to CSD. It also shows that the same CAR construct can elicit different effects between Tregs and Tconv.

To date, CAR-T cell manufacturing according to GMP requirements includes polyclonal stimulation (CD3/CD28). This expansion step leads to a cell product whose function could be significantly altered by CAR tonic signaling. Hence, the effect of constitutive CAR activation on Treg biology and fate during TCR-driven CAR-Treg expansion is a critical question on the path toward clinical translation.

Furthermore, we believe that our data may shed light on the mechanisms underpinning CAR-Treg dysfunction in other reports. For instance, a number of clues suggest that the dramatic lack of suppressive function of highly proliferative TNFR2 CAR-Tregs could result from CAR tonic signaling (Dawson Sci Transl med 2020). In fact, baseline expression of activation markers (CD71 and LAP), heightened proliferation, and increased inflammatory cytokine production upon CAR-independent stimulation, in comparison to their CD28 counterparts, are together consistent with CAR tonic signaling. Whether mitigation of TNFR2-related tonic signaling improves TNFR2 CAR-Treg function has yet to be investigated.

With respect to the novelty of the study, six points could be highlighted:

> 1- We compared two HLA-A2-targeted CARs sharing a similar architecture (ScFv, hinge, ...), with the exception of CSD. Notably, both CD28- and 4-1BB- CARs demonstrated evidence of tonic signaling based on phenotypic, signaling, metabolic, and transcriptomic studies. However, our study reveals that the impact of tonic signaling on Treg biology, function and longevity varies greatly according to CAR CSD.

Page 20: "In this study, we observed ... constructs with different ensuing biological effects."

Page 24: "These data highlight the ... constitutive 4-1BB signaling."

Pages 23-24: "It was also proposed that the close proximity ... a 3rd generation CAR²¹."

> 2- Our study shows that the impact of 4-1BB tonic signaling varied between T cell lineages (Treg vs Tconv), despite similar CAR transduction efficiency. More specifically, these differences included the following:

- The metabolic switch induced by 4-1BB tonic signaling dramatically differs between Tregs and Tconvs. This finding is in line with enhanced mTORC1 activation in 4-1BB-CAR-Tregs but not in 4-1BB-CAR-Tconvs compared to their CD28-CAR and untransduced counterparts.
- With our construct, 4-1BB tonic signaling enhanced TCR-driven proliferation and lengthened the blastic phase in CAR-Tregs, unlike in CAR-Tconvs.
- Induction of HLA-DR expression, a hallmark of the 4-1BB tonic signal (Boroughs Mol Ther 2020), was more important in CAR-Tregs than in CAR-Tconvs
- 4-1BB tonic signaling destabilized CAR-Treg stability and function, whereas it was previously shown to spare antitumor efficacy in CAR-Tconvs (Milone Mol Therap 2009).

Pages 21-22. "In addition, our study indicates ... CAR-T cells⁵¹."

>3- Our study shows that the failure of 4-1BB CAR-Tregs to induce early activation markers in response to antigen receptor stimulation can be progressively acquired over time in cell culture. This indicates that the reduced capacity of CAR-Tregs to be activated through the CAR can be indicative of tonic signaling rather than of a defective CAR construct.

>4- On the other hand, potent 4-1BB tonic signaling is not necessarily associated with increased baseline expression of CD69. The hints indicate that tonic signaling could be very subtle. For instance, in our study, CD28 CAR tonic signaling was revealed by slight changes in the transcriptomic profile (IL10), recruitment of signaling phosphoproteins, and hastened loss of Helios expression among FOXP3+ cells. Systematic screening for CAR tonic signaling, which could otherwise be easily overlooked, is thus important for fully assessing the impact of CSD on CAR-Tregs. We thus believe that thorough study of the different forms of tonic signaling, according to CSD, will provide a more complex, yet accurate, picture of their effect in CAR-Tregs.

>5- This study, along with another one from Vancouver's group, available as a preprint (Lamarque BioRxiv 2020), highlights the pleiotropic forms of tonic signaling in CAR-Tregs, as already well established in CAR-Tconvs. This point is further discussed in the revised manuscript.

Pages 25: "Regarding Tregs, a preprint from Vancouver's group ... despite Treg lineage stability⁵⁹."

>6- Our study demonstrates that pretreatment with mTOR inhibitor/vitamin C significantly rescues 4-1BB CAR-Tregs from tonic signal-induced dysfunction.

(2). It might be argued that novelty of the study relates to the identification of tonic signaling in a CAR Treg context. However, I also consider this incremental given the fact that tonic signaling in a well recognised phenomenon in Tconv T-cells. Moreover, tonic signaling mediated by 4-1BB-containing second generation CARs has been described in Tconv cells previously by Milone et al (ref 44 in the manuscript) and also by Gomes-Silva et al (ref 26 in the manuscript). Outcomes were markedly different in these two examples such that activated CAR T-cell proliferated constitutively in the former example, while activation induced cell death occurred in the latter case. Here, we see a broadly similar form of tonic signaling to that described by Milone.

We have not sufficiently emphasized the differences and specificities relative to Tregs. Our results precisely show that the impact of 4-1BB tonic signaling differs between Tregs and Tconvs, as described in a previous study. Therefore, we believe that the conclusions drawn in Tconv studies, with respect to 4-1BB tonic signaling should not be taken for granted in Tregs.

With respect to the abovementioned papers:

1- Gomes-Silva et al.: This study compellingly demonstrated that 4-1BB signaling-associated toxicity in CAR T cells resulted from a TRAF2-dependent positive feedback loop that enhanced LTR promoter-driven CAR expression in a nonself-inactivating gammaretroviral vector. In contrast, the authors show that this negative effect was dramatically reduced when CAR expression was driven by an EF1a promoter in a self-inactivating lentiviral vector, such as the one used in our study.

2- Milone et al.: As mentioned by Reviewer 2, this study showed an increased *in vitro* proliferation rate of 4-1BB CAR T cells when stimulated independently of CAR ligation, along with a more sustained blast phase. We agree with Reviewer 2 that our finding in Tregs is reminiscent of this seminal study. However, this paper also demonstrates that 4-1BB-CAR T cells exhibit greater *in vivo* survival. The devastating effect of 4-1BB tonic signaling on Treg stability, function and survival is unique to this lineage.

(3). Specifically, Milone et al observed CAR ligand-independent proliferation of CD19-specific CAR T-cells (presumed Tconv) of both the CD4+ and CD8+ T cell lineage. This was "associated with a prolonged blast phase after the initial stimulation and transduction". Milone concluded that "the enhanced growth effects of the α CD19-BB- ζ receptor are consistent with the antigen-independent growth effects that are observed in T cells stimulated through the natural CD137 receptor by agonist monoclonal antibody." These results are broadly similar to those reported here in Tregs. It would be useful to clarify if this effect is dependent on recent activation through the TCR? (as occurs on d11 according to the scheme shown in Fig. 1c).

We agree with Reviewer 2. However, our results show that 4-1BB tonic signaling seems detrimental to CAR-Treg function, whereas Milone et al. reported that CAR-Tconv function was spared by 4-1BB tonic signaling. Whether this difference results from a greater interaction between the CAR and a highly constitutive TCR signal in Tregs (Jennings E Cell Reports 2020) is an interesting hypothesis that has yet to be explored.

We also further explored the interaction between 4-1BB tonic signaling and TCR activation.

Pages 18-19: "4-1BB tonic signaling ... counterparts."

This additional experiment suggests that maintenance of 4-1BB tonic signaling is not dependent on sustained TCR-induced activation. However, we cannot exclude that its initiation requires activation through the TCR.

(4). The introductory section over-simplifies what is known about tonic signaling. Multiple forms of tonic signaling have been described in Tconv T-cells engineered to express either CD28 or 4-1BB containing CARs. However, the authors preferentially link tonic signaling to CD28-containing CARs with statements in the Introduction (first two) and Discussion (third statement) such as

- "Tonic signaling may cause CAR-T cell dysfunction, especially when the CAR structure incorporates the CD28 costimulatory domain (CSD) rather than the 4-1BB CSD."
- "This study demonstrates that CD28-CAR best preserves Treg function and survival in the context of tonic signaling, in contrast with previous findings for Tconvs."
- "Strikingly, although CD28 was found to worsen the process of tonic signaling-induced exhaustion in Tconvs, our data showed the opposite in Tregs."

The undue emphasis on CD28-associated tonic signaling seems to imply that the demonstration here of 4-1BB tonic signaling in a Treg is a highly novel finding rather than something that might be expected, given the full breadth of the prior literature.

A more general picture of the complexity of CAR tonic signaling has been included in the revised manuscript. We also better emphasize the issues related to tonic signaling specifically in Tregs. The respective role of CD28 and 4-1BB CSD in Tconvs and Tregs has been moved to the discussion section to avoid an overly long introduction.

Pages 5-6: (introduction) "CAR tonic signaling may ... according to the type of CSD."

Page 20. (discussion) "With respect to tonic signaling ... *in vivo* efficacy of CAR-T cells⁵¹."

(5). Tonic signaling was not characterised by unregulated expression of CD69 or constitutive cytokine production. Why is only HLA-DR upregulated? Did authors test for tonic signaling using other assays (eg constitutive phosphorylation of CD3z; Jurkat Nur77-RFP reporter cells (Smith et al Sci Transl Med 2019: Vol. 11, Issue 485, eaau7746)).

A recent study (Boroughs Mol Ther 2020), based on single-cell and bulk transcriptomic analyses, disclosed specific signatures related to CD3 ζ and 41BB CSD tonic signals in conventional CD4 and CD8 T cells. Interestingly, 4-1BB tonic signaling included high expression of HLA class II, in keeping with our finding in Tregs. In our study, evidence for 4-1BB tonic signaling was plentiful and based on differences in proliferation rate, metabolic

switch, phenotypic changes, transcriptomic profile, and signaling pathways observed in 4-1BB CAR-Tregs that were not stimulated through the CAR.

Page 21 (discussion) “A recent report described a ... tonic signal in Tconv³⁶.”

(6). Authors comment on the affinity of scFvs but no formal affinity data are presented (e.g. Kd, on rates, off rates etc).

We agree that the Luminex assay only provides a semiquantitative assessment of the binding strength and more compelling information about antigen specificity. Therefore, we discarded the word “affinity” and only emphasized the ability to investigate antigen specificity.

Page 7 (Results): “To assess their specificity, ... class I antigen (Supp Figure 1a).”

Page 27 (Methods section). “**scFv specificity:** The specificity ... (LABScreen® Single Antigen).”

(7). Authors demonstrate higher expression of 4-1BB CAR than CD28 CAR, indicated by higher pentamer and protein L binding. Is this due to enhanced transcription, given the link between 4-1BB tonic signaling and promoter selection (Gomes-Silva reference)?

We do not think that our results support enhanced transcription of the CAR.

First, in the above-cited Gomes-Silva reference, 4-1BB tonic signaling increases CAR transcription through NFκB-dependent activation of the nonself-inactivating LTR that drives expression of the transgene. The enhanced CAR expression and ensuing deleterious effects were abolished when a self-inactivating lentiviral vector was used, as in our study.

Second, if CAR expression was increased by 4-1BB tonic signaling, EGFR expression would be similarly increased in 4-1BB CAR-Tregs. In fact, we agree with Reviewer #2’s next comment: “Coexpression of CAR and EGFRt reporter should theoretically be stoichiometric given the use of a T2A peptide”. However, EGFRt was not differentially expressed across the different CAR and T cell lineages.

Third, and most importantly, quantification of CAR mRNA using two different sets of primers did not display any difference between the two CAR constructs.

Page 11: “In fact, levels of CD28- and 4-1BB-CAR-T2A-EGFRt mRNA ... by RT-PCR (Figure 2i).”

(8). Co-expression of CAR and EGFRt reporter should theoretically be stoichiometric given the use of a 2A peptide. In keeping with this, authors state that “the mean fluorescence intensities (MFIs) of HLA-A2 pentamer and EGFR staining were not different across the CAR CSDs and cell lineages (Figure 2D).” To this reviewer however, there appears to be a trend towards higher expression of the 4-1BB CAR (indicated by higher pentamer and protein L binding), accompanied by a trend towards reduced EGFRt expression by 4-1BB CAR Tregs (all shown in Fig. 2d). Further replicate experiments and expression of the data as a ratio between CAR expression to EGFRt expression might unmask a significant difference in this respect. This underpins the need for transcriptional data.

Thank you for this terrific suggestion. Transcriptomic analysis was performed, and the point raised by Reviewer 2 is further discussed.

Page 11: “Since the flow cytometry ... expression at the cell membrane.”

(9). Authors infer that 4-1BB tonic signaling was primarily seen in Tregs. How was tonic signaling excluded in Tconv that expressed these CARs.

We apologize if our previous statement was misleading. We did not mean that 4-1BB CAR-Tconvs were devoid of tonic signaling. In fact, Fig. 2a shows a trend toward greater baseline HLA-DR expression in 4-1BB-CAR-Tconvs than in CD28-CAR-Tconvs. However, we indicated that the effect of 4-1BB tonic signaling induced by the same CAR construct differed sharply between Tregs and Tconvs and was far more pronounced in Tregs.

(10). What is the evidence that MAPK activation drives 4-1BB mediated proliferation.

We did not want to note that MAPK activation was the primary driver of sustained 4-1BB-CAR-Treg proliferation. We stressed the importance of tight control of the MAPK pathway in Tregs and drew readers' attention to the possibility that a dysregulated MAPK pathway could contribute to destabilizing CAR-Tregs.

Page 23: "Similarly, DUSP4-dependent ... and impede Treg function."

(11). I may have missed this but I could not identify the supplementary Tables.

We apologize for this omission. The supplementary tables are presented in the revised manuscript (Supplementary Material).

(12). While the Dawson citation compromises novelty, a number of differences emerge between the two studies. For example, Dawson reported preservation of high levels of central memory Treg by 4-1BB CAR Tregs whereas effector differentiation was more evident here. Please discuss differences in findings regarding 4-1BB CAR Tregs between the two studies.

We are grateful to Reviewer 2 for pointing out this interesting issue, which is now discussed as follows:

Pages 22-23: Tonic signal-induced ... constitutive CAR activation.

(13). The analysis of immunometabolism is cursory, incorporating only a limited number of genes.

Metabolic profiling was added to the revised manuscript and further demonstrated that the impact of CAR tonic signaling on metabolism depends on both the T cell lineage (Treg vs Tconv) and CSD.

Pages 9-10: "In contrast to ... in 4-1BB CAR-Tregs."

(14). What were the kinetics of kinetics of s6 phosphorylation in 4-1BB CAR Tregs after activation on d11?

We agree that the kinetics of pS6 phosphorylation would be useful to capture the timing of 4-1BB-induced mTOR activation and possible defective dephosphorylation pathways. However, within the time frame dedicated to the revision, we had to prioritize other experiments, including metabolic and transcriptomic analyses, as well as an *in vivo* model with untreated 4-1BB CAR-Tregs.

(15). It is stated that "In line with these findings, the expansion rate of 4-1BB CAR-Tregs decreased to a level comparable to those of CD28 CAR-Tregs and untransduced Tregs (Figure 5D)." However, I do not see the CD28 and untransduced Treg controls in Figure 5D.

We have fixed the figure accordingly.

(16). Could *in vivo* persistence of Tregs in the present study be linked to the use of a different conditioning regimen to that used in other studies?

We cannot address this question given the lack of irradiators at our animal house facility.

Alternatively, but not exclusively, we think that administration of CAR-Tregs and their target cells through different IV routes might at least in part account for longer *in vivo* persistence. In fact, *Cazaux M et al.* previously showed that the binding of CD19-specific CAR-T cells to their CD19+ target cells immediately after their administration strongly limits their ability to recirculate through lymphoid organs. This finding raised the concern that mixing HLA A2-targeted CAR-Tregs and HLA A2+ PBMCs before infusion would significantly impede their circulation. Hence, we performed two preliminary experiments (a total of 5 mice in each group), where CAR-Tregs and PBMCs were administered either separately or together. Based on the results depicted in the figure below, we opted for separate administration, even though the difference fell short of statistical significance, likely due to low sampling size.

(17). Why is pulmonary transit of Treg not evident in figure 7?

Our experiments were not designed to investigate the early pulmonary transit of CAR-Tregs. In the above-cited paper (Cazaux et al. paper), cell clusters formed by CAR-T cells bound to their target cells were trapped in the lungs as early as 15 minutes after cell transfer. In our experiments, a bioluminescent signal was still readily detected in some animals at early time points (day 5).

Figure 7 has been changed, as we feel that individual signal trajectories would be more compelling than selected pictures.

(18). Authors comment that their findings conflict with two recent studies reporting the lack of *in vivo* suppressive function of 4-1BB CAR-Tregs. Could this be due to the fact that the Treg-supporting cocktail was added to the culture medium from days 10 to 18 for 4-1BB CAR-Tregs prior to *in vivo* testing?

This is a well-taken point. We have addressed this question with another group of mice treated with 4-1BB CAR-Tregs cultured in the absence of a Treg-friendly cocktail. 4-1BB CAR-Tregs not only failed to control GVHD but also boosted human PBMC early expansion and soon vanished from the circulation.

The dramatic difference in terms of *in vivo* suppressive capacities and survival between untreated and pretreated 4-1BB CAR-Tregs further demonstrates the impact of tonic signaling during CAR-Treg manufacturing.

Pages 16: “Regarding survival, ... in controls (Figure 6e).”

Page 23: “More importantly, ... multikinase inhibition⁵⁸.”

(19.) Authors comment that “4-1BB CAR-Tregs potently suppressed xenoGVHD despite a reduced ability to be stimulated through the CAR. This finding is reminiscent of a previous report showing *in vivo* antigen-specific suppression mediated by HLA-A2-targeted CAR-Tregs, even in the absence of signaling domains in the CAR structure.” Is this not likely to be due to the ability of signaling defect CARs to facilitate retention of these Tregs at the site of the target antigen (HLA-A2).

We have toned down this point in the revised manuscript. Although 4-1BB CAR-Tregs collected after two weeks of culture consistently failed to induce CD69 upon CAR stimulation, they still demonstrated the ability to upregulate CD25 and 4-1BB. Hence, 4-1BB CAR-Tregs demonstrate a reduced but not abolished ability to be stimulated through CAR.

Pages 17-18: “4-1BB tonic signaling reduces the ability of CAR-Tregs to be stimulated.

We wondered whether 4-1BB-*in vivo* signal ... CAR stimulation. (Figure 8a).”

Minor points

(1.) It is incorrect to state that CARs are engineered receptors that consist of an extracellular single-chain variable fragment (scFv).... Many CARs employ alternative ligand binding moieties to scFvs.

The sentence has been rephrased as follows:

Page 4: “The antigen-binding motif is frequently a single-chain variable fragment (scFv) which combines the heavy and light chains of an antibody⁷.”

(2.) In the cartoons shown in Fig 1a, it would be helpful to include additional structural details of CARs eg hinge/ spacer and transmembrane domain.

Revised Fig 1a includes additional details about the CAR structure.

(3.) A reference citation should be provided for EGFRt (Jensen et al).

Page 7: This reference has been added, as suggested.

(4.) Please specify the promoter used in the LV vector, given the known importance of promoter choice in 4-1BB tonic signaling (Gomes-Silva reference).

We apologize for this omission. The information is now provided in the results section:

Page 7: “Both bicistronic constructs were next incorporated into a pCCL self-inactivating lentivirus vector (LV) behind an EF-1 alpha promoter (Figure Supp 1c).”

(5.) A dotted line symbol should be included in the legend for Fig 1e (NT cells).

The dotted line symbol has been included in the revised legend.

(6.) It is preferable not to use red and green colors in figures, in consideration of color-blind individuals.

We sincerely apologize for the lack of consideration for color-blind individuals. The colors have been changed to a color blind-friendly chart.

Reviewer #3 (Remarks to the Author):

The authors of this manuscript aimed to study the tonic signalling in Tregs transduced with CAR constructs. They compared Tregs transduced with two CAR constructs both specific for the same antigen (HLA-A2) but with two different costimulatory endodomains. They demonstrated that the Tregs expressing the CAR with the 4-1BB, compared to CD28, proliferated more, had greater activation of MAP kinase and mTOR pathways, but decreased stability and capacity to be restimulated via the CAR.

This paper addresses a novel question, namely the effect of the costimulatory endodomains in the tonic signal in Tregs. This question has been addressed before with conventional T cells. Altogether this is an interesting and well performed study showing very clearly that for Tregs CD28 reduced tonic signalling compared to 4-1BB, opposite of what was seen for conventional T cells. However, I have a series of comments that are listed below.

1.) One reservation that I have is that the authors did not compare their results with what was published by Boroughs et al in 2019 in JCI Insight. Although the aims of the two studies were different some analysis of the Tregs are in common but the results are different. The authors need to add this reference and compare their data with the results from Boroughs study.

In the revised manuscript, our results are now better discussed in light of previous conflicting results. More specifically, we believe that the discrepancies can be explained by the following:

- In our study, the “naïve” origin of Tregs might account for the ability of 4-1BB CAR-Tregs to maintain hallmark features of Tregs, unlike in the report from Boroughs et al.;
- The addition of a Treg-friendly cocktail definitively mitigated the negative effect of 4-1BB tonic signaling on CAR-Treg function.

Page 23: “Together, these results ... multikinase inhibition⁵⁸.”

2.) Linked to the previous point in the Introduction (line 107) the authors need to cite other publications where the negative effect of 4-1BB was explored for Tregs.

We did not attempt to be exhaustive with regard to the impact of endogenous 4-1BB signaling and Tregs for two main reasons:

- Our manuscript includes 72 references, a number already far above the standards for an original article.
- A recent publication from Boroughs et al. 2020 precisely demonstrated that the transcriptomic signatures induced by 4-1BB CAR CSD and endogenous 4-1BB signaling were not similar, although they shared common features. Therefore, we paid more attention to the publications that addressed the effect of 4-1BB CSD in CAR-Tregs (Boroughs JCI Insight 2019, Dawson STM 2020, Nowak Front Immunol 2018).

3.) The authors need to clarify in the text the way that the Tregs were stimulated for each result reported. For example, in Figure 1e the authors wrote in the Figure legend that the Tregs were 'TCR restimulated'. Is this an allogeneic stimulation of splenocytes expressing HLA-A2? If this is the case both TCR and CAR are engaged, is this correct? The authors need to clarify this point and where possible separate the TCR stimulation from the CAR. The authors should at least comment on this point.

We apologize for the confusion. The ex vivo expansion protocol was based on two rounds of aCD3/aCD28 bead stimulation, as indicated in Figure 1c and Figure 8b. TCR restimulation refers to the second activation with anti-CD3/CD28 beads, either on day 11 (Figure 1c) or day 8 (Figure 8b). Therefore, most of the experiments were based on these CAR-independent culture protocols that allowed us to assess ligand-independent CAR tonic signals.

However, there are few exceptions for experiments designed to assess *in vitro* activation through CAR ligation.

- In Figure 1d, CAR-Tregs were stimulated on day 10 in an HLA A2-specific manner before the second round of aCD3/aCD28 stimulation. As stimulators, HLA A2+ or HLA A2- splenocytes were used and compared to anti-CD3/CD28 beads.
- In Figure 8, CAR-Tregs were also stimulated in an HLA A2-specific manner either at day 14 or 16, after separation from the beads. As stimulators, HLA A2+ pentamers or HLA A2+ cell lines were used and compared to controls.

The legends have been updated to avoid any ambiguity.

4.) The authors speculate on lines 169-171 that 'ligand-independent CAR tonic signalling seems to primarily dependent on the high cell-surface density of the CAR', which altogether entail CAR clustering'. The authors speculate that the high expression density of 41BB seen with pentamers correspond to a cluster of CAR molecules. However, this should be demonstrated directly using other approaches such as confocal microscopy. In addition, the authors say that proliferation is induced by endogenous TCR, and linked to the previous point does this mean that they used splenocytes HLA-A2 negative?

As discussed above, the Penn group previously demonstrated that incorporation of 4-1BB CSD induced enhanced proliferation and a more prolonged blastic phase through ligand-independent tonic signaling (Milone et al. 2009). This finding, referred to as 4-1BB tonic signaling, was not observed with either the 28 ζ or third-generation 28BB ζ counterparts, which otherwise shared the same CAR structure. Together, these data suggest that ScFv oligomerization/CAR clustering may not play an important role in the mechanisms underpinning such 4-1BB tonic signaling (Ajina et al. 2018). Instead, this effect seems to depend upon the relative position of 4-1BB CSD from the cell membrane. Hence, we did not further explore the ability of our HLA A2-targeted CAR to self-aggregate.

5.) The author report on lines 172-175 that 'the vector copy number (VCN) in transduced Tregs was similar between CD28 CAR and 4-1BB CAR'. However, the VCN is not enough to demonstrate that the number of CD28 and 41BB molecules are the same. For instance, the mRNA of 41BB might be more stable. The level of total 41BB or CD28 protein should be evaluated by western blot or other approaches to quantify the amount of target protein normalised to the total cell proteins.

6.) The authors need to modify their statement in line 176 as it is not fully correct. Data only show a higher expression of 41BB molecule on the cell surface and not necessarily a greater vector expression (VCN similar) and higher protein production (see comment above).

7.) On line 188-191, the authors contradict themselves. On line 169 they suggest that the tonic signal of 41BB is due to the higher expression of CAR molecule. However, using protein L they show CAR expression is similar in both in CD28 and 41BB CAR cells. This point needs to be clarify. The statement "these results suggest that constitutive 41BB stimulation itself interferes with human Treg biology" is not clear and should be explained more in detail.

We have addressed these important points using CAR mRNA quantification instead of western blotting, which requires a much greater number of cells. Notably, the levels of CD28 and 4-1BB CAR mRNA were strictly similar, irrespective of the set of primers used to amplify either the ScFv or the EGFRt part of the mRNA.

This paragraph has been rephrased as follows:

Pages 10-11: "4-1BB tonic signaling in Tregs does not result ... the cell membrane."

8.) The phosphorylation of the MAP kinase and AKT pathways has been observed after 5 days of stimulation with beads from what is shown in Figure 1c (is this correct)? What happen after few more days? Is the phosphorylation maintained at the same level or decreases?

We agree that the kinetics of S6, Akt, and ERK1/2 phosphorylation would be useful for capturing the timing of 4-1BB-induced mTOR activation and possible defective dephosphorylation pathways. However, given the number of additional experiments we wished to perform within the time frame dedicated to the revision, we had to prioritize other experiments, including metabolic and transcriptomic analyses, as well as an *in vivo* model with untreated 4-1BB CAR-Tregs.

9.) In the p-tyrosine blot (Fig 2 e), did the authors investigate the differential expression of tyrosine phosphorylation in 41BB and CD28 CAR Treg cells at just below 125 (41BB) or above 50 (CD28) kDa? That seems to be particularly relevant to understand the different signals induced by the two CSDs.

This important field of investigation is an upcoming project, currently subject to a grant proposal. We will evaluate the signaling and transcriptional networks triggered by CARs with 4-1BB, CD27 and CD28 CSD in detail when expressed in Treg vs Tconv cells using interactome analysis, phosphoproteome analysis and single-cell analysis. These cell-consuming and costly investigations, which require experimental fine-tuning, could unfortunately not be performed for the revision of this manuscript.

10.) The CD28 and 41BB CAR Treg cells in the tSNE analysis do not seems highly clustered as stated by the

authors (line 225-228). Although there is a clearly different clusterisations, all the markers shown in Fig. 3a, with the exception of HLA-DR and CD15s, seem to form clusters in both populations. The authors should also comment about the over-expression of 41BB in 41BB CAR Treg cells. Can the CAR signal self-support its own expression? The authors should also stress that the tSNE analysis shows difference in the expression and not necessarily identify positive and negative cell populations.

A tSNE algorithm does not preserve distances or density. With that in mind, we prefer to remain cautious when discussing the clustering of Treg subsets. We used t-SNE analysis primarily to visualize that the two CAR-Treg populations were dramatically separated from one another, likely due to the sharp difference in HLA-DR expression. This analysis also helped us to delineate a subset of 4-1BB CAR-Tregs that fulfilled all the phenotypic hallmarks of effector Tregs. In addition, we noticed that CAR-CD28 Tregs maintained greater CCR7 expression than 41BB-CAR-Tregs. This unexpected finding is now further discussed:

Pages 22: “The transcription factor MYB, ... Day 16 of culture⁴⁰.”

Pages 22-23: “Tonic signal-induced sustained Akt/... constitutive CAR activation.”

11.) The production of cytokines in Figure 3b shows a difference with the study from Boroughs for TNF- α , the authors need to comment on this.

As discussed above, we believe that the origin of the cells (CD45RA+ CD45RO- CD25+ CD127-naïve Tregs in our study vs total CD25+ CD127- Tregs in Borough’s study) could account for the greater ability of our 4-1BB-CAR-Tregs to preserve the epigenetic and transcriptomic Treg programs despite 4-1BB tonic signaling. However, it is important to stress that our 4-1BB CAR-Tregs with constitutive activation were highly dysfunctional, in line with previous studies, whereas those pretreated with an mTOR inhibitor and vitamin C recovered suppressive capacities *in vivo*.

12.) In Figure 5, the authors report the culturing the 41BB Tregs with Rapa/ VitC on different parameters. However, the authors should present additional data presented in Figure 1-4 before using the Rapa/VitC *in vivo*.

Figure 5 has been changed accordingly. Panels b and c, which depict the *ex vivo* fold expansion and the frequency of double-negative FOXP3 HELIOS, respectively, show the CD28 group for comparison.

13.) For the experiment *in vivo*, I do not understand why the authors have tested only the 41BB Tregs treated with Rapa/VitC as the entire aim of the study was to compare the two co-stimulatory endo-domains in the absence of any additional factors that can modify the behaviour of the Tregs. In my view (although I realise it is a lot of work) these two Treg lines (in the absence of Rapa/VitC) should be compared.

Thank you for suggesting this additional experiment, which has definitively strengthened our study.

We included another group of mice treated with 4-1BB CAR-Tregs cultured in the absence of a Treg-friendly cocktail. Comparison between untreated and pretreated 4-1BB CAR Treg populations allowed us to assess the beneficial effect of transient exposure to the Rapa/vit C cocktail in rescuing *in vivo* suppressive function and *in vivo* expansion (Revised Figures 6 and 7). Notably, 4-1BB CAR-Tregs not only failed to control GVHD but also boosted human PBMC early expansion and soon vanished from the circulation.

The dramatic difference in terms of *in vivo* suppressive capacities between untreated and pretreated 4-1BB CAR-Tregs further demonstrates the impact of tonic signaling *ex vivo* CAR-Treg expansion.

Pages 16: “Regarding survival, ... in controls (Figure 6e).”

Page 23: “More importantly, the addition ... multikinase inhibition⁵⁸.”

14.) It is not clear to me but I imagine that the 41BB Tregs in Figure 8 were not treated with Rapa/VitC. If this is correct more reason for the 41BB Tregs in the *in vivo* experiment to be untreated. At least the author should compare in the experiment described in Figure 8 the 41BB Tregs treated and not with Rapa/VitC.

We apologize if the legends were not clear enough, but the two populations are depicted: 4-1BB CAR-Tregs with (yellow) and without (red) Rapamycin.

Minor

1.) Can the authors describe in the MM what the HLA-A2/A28-negative cytopheresis kits is?

Page 8: We apologize for this misleading sentence. This is a bad translation. One must read:

“... CD4+ CD25- CD127+ CD45RA+ CD45RO- ... HLA-A2/A28-negative donors (Figure 1b).”

2.) There is a mistake on line 143 the authors wrote HLA-A28 negative, should be HLA-A2 negative.

Page 8: We apologize for the typo. One should read “HLA-A2/A28 negative”.

REVIEWER COMMENTS

Reviewer #1 (Remarks to the Author):

The authors have addressed all my comments and suggestion raised to the initial version of the manuscript. I have no additional comments.

Reviewer #2 (Remarks to the Author):

I believe that the authors have satisfactorily addressed the technical points raised in my review.

Reviewer #3 (Remarks to the Author):

The authors have replied to my comments in a satisfactory manner.